# Can machine learning improve the model representation of TKE dissipation rate in the boundary layer for complex terrain?

Nicola Bodini[1,2], Julie K. Lundquist[1,2], and Mike Optis[2]

[1]Department of Atmospheric and Oceanic Sciences, University of Colorado Boulder, Boulder, Colorado, USA
[2]National Renewable Energy Laboratory, Golden, Colorado, USA

**Correspondence:** Nicola Bodini (nicola.bodini@nrel.gov)

**Abstract.** Current turbulence parameterizations in numerical weather prediction models at the mesoscale assume a local equilibrium between production and dissipation of turbulence. As this assumption does not hold at fine horizontal resolutions, improved ways to represent turbulent kinetic energy (TKE) dissipation rate ($\epsilon$) are needed. Here, we use a 6-week data set of turbulence measurements from 184 sonic anemometers in complex terrain at the Perdigão field campaign to suggest improved representations of dissipation rate. First, we demonstrate that the widely used Mellor, Yamada, Nakanishi, and Niino (MYNN) parameterization of TKE dissipation rate leads to a large inaccuracy and bias in the representation of $\epsilon$. Next, we assess the potential of machine-learning techniques to predict TKE dissipation rate from a set of atmospheric and terrain-related features. We train and test several machine-learning algorithms using the data at Perdigão, and we find that the models eliminate the bias MYNN currently shows in representing $\epsilon$, while also reducing the average error by up to almost 40%. Of all the variables included in the algorithms, TKE is the variable responsible for most of the variability of $\epsilon$, and a strong positive correlation exists between the two. These results suggest further consideration of machine-learning techniques to enhance parameterizations of turbulence in numerical weather prediction models.

*Copyright statement.* This work was authored in part by the National Renewable Energy Laboratory, operated by Alliance for Sustainable Energy, LLC, for the U.S. Department of Energy (DOE) under Contract No. DE-AC36-08GO28308. Funding provided by the U.S. Department of Energy Office of Energy Efficiency and Renewable Energy Wind Energy Technologies Office. The views expressed in the article do not necessarily represent the views of the DOE or the U.S. Government. The U.S. Government retains and the publisher, by accepting the article for publication, acknowledges that the U.S. Government retains a nonexclusive, paid-up, irrevocable, worldwide license to publish or reproduce the published form of this work, or allow others to do so, for U.S. Government purposes.

## 1 Introduction

While turbulence is an essential quantity that regulates many phenomena in the atmospheric boundary layer (Garratt, 1994), numerical weather prediction models are not capable of fully resolving it. Instead, they rely on parameterizations to represent some of the turbulent processes. Investigations into model sensitivity have shown that out of the various parameterizations

currently used in mesoscale models, that of turbulent kinetic energy (TKE) dissipation rate ($\epsilon$) has the greatest impact on the accuracy of model predictions of wind speed at wind turbine hub height (Yang et al., 2017; Berg et al., 2018).

Current boundary layer parameterizations of $\epsilon$ in mesoscale models assume a local equilibrium between production and dissipation of TKE. While this assumption is generally valid for homogeneous and stationary flow (Albertson et al., 1997), as the horizontal grid resolution of mesoscale models is constantly pushed toward finer scales thanks to the increase of the computing resource capabilities, the theoretical bases of this assumption are violated. In fact, turbulence produced within a model grid cell can be advected farther downstream in a different grid cell before being dissipated (Nakanishi and Niino, 2006;

Krishnamurthy et al., 2011; Hong and Dudhia, 2012).

The inaccuracy of the mesoscale model representation of $\epsilon$ impacts a wide variety of processes that are controlled by the TKE dissipation rate. In fact, the dissipation of turbulence affects the development and propagation of forest fires (Coen et al., 2013), it has consequences on aviation meteorology and potential aviation accidents (Gerz et al., 2005; Thobois et al., 2015), it regulates the dispersion of pollutants in the boundary layer (Huang et al., 2013), and it affects wind energy applications (Kelley

et al., 2006): for example, in terms of the development and erosion of wind turbine wakes (Bodini et al., 2017).

Several studies have documented the variability of $\epsilon$ using observations from both in-situ (Champagne et al., 1977; Oncley et al., 1996; Frehlich et al., 2006) and remote-sensing instruments (Frehlich, 1994; Smalikho, 1995; Shaw and LeMone, 2003). Bodini et al. (2018, 2019b) showed how $\epsilon$ has strong diurnal and annual cycles onshore, with topography playing a key role in triggering its variability. On the other hand, offshore turbulence regimes (Bodini et al., 2019a) are characterized by smaller

values of $\epsilon$, with cycles mostly impacted by wind regimes rather than convective effects. Also, $\epsilon$ greatly increases in the wakes of obstacles, for example wind turbines (Lundquist and Bariteau, 2015; Wildmann et al., 2019) or whole wind farms (Bodini et al., 2019b).

This knowledge on the variability of TKE dissipation rate provided by observations lays the foundation to explore innovative ways to improve the model representation of $\epsilon$ in the atmospheric boundary layer. In this study, we leverage the potential of

machine-learning techniques to explore their potential application to improve the parameterizations of $\epsilon$. Machine-learning techniques can successfully capture the complex and nonlinear relationship between multiple variables without the need of representing the physical process that governs this relationship. They have been successfully used to advance the understanding of several atmospheric processes, such as convection (Gentine et al., 2018), turbulent fluxes (Leufen and Schädler, 2018), and precipitation nowcasting (Xingjian et al., 2015). The renewable energy sector has also experienced various applications of

machine-learning techniques, in both solar (Sharma et al., 2011; Cervone et al., 2017) and wind (Giebel et al., 2011; Optis and Perr-Sauer, 2019) power forecasting. Applications have also been explored at the wind turbine level, for turbine power curve modeling (Clifton et al., 2013), turbine faults and controls (Leahy et al., 2016), and turbine blade management (Arcos Jiménez et al., 2018).

Here, we train and test different machine-learning algorithms to predict $\epsilon$ from a set of atmospheric and topographic vari-

ables. Section 2 describes the Perdigão field campaign and how we retrieved $\epsilon$ from the sonic anemometers on the meteorological towers. In Section 3, we then evaluate the accuracy of one of the most common planetary boundary layer parameterization schemes used in numerical weather prediction: the Mellor, Yamada, Nakanishi, and Niino (MYNN) parameterization scheme

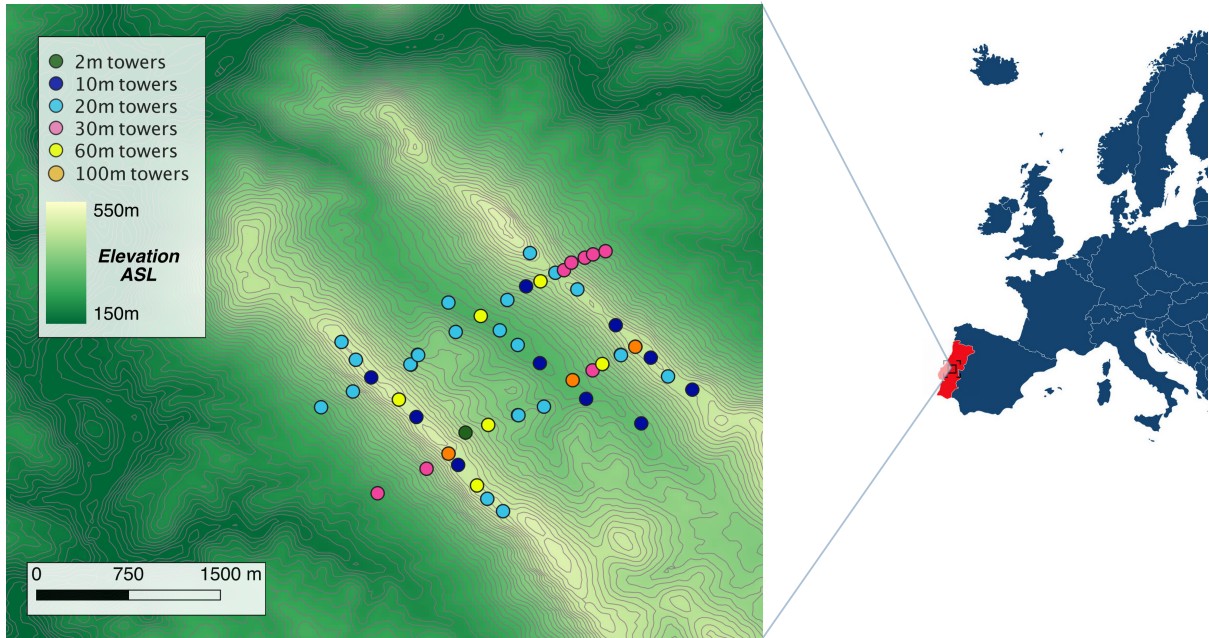

**Figure 1.** Map of the Perdigão valley showing the location and height of the 48 meteorological towers whose data are used in this study. Digital elevation model data courtesy of the U.S. Geological Survey.

(Nakanish, 2001). Section 4 presents the machine-learning algorithms that we used in our analysis. The results of our study are shown in Section 5, and discussed in Section 6, where future work is also suggested.

## 2   Data

### 2.1   Meteorological towers at the Perdigão field campaign

The Perdigão field campaign (Fernando et al., 2018), an international cooperation between several universities and research institutes, brought an impressive number of instruments to a valley in central Portugal to survey the atmospheric boundary layer in complex terrain. The Perdigão valley is limited by two mountain ridges running from northwest to southeast (Figure 1), separated by ∼1.5 km. The intensive operation period (IOP) of the campaign, used for this study, was from 1 May to 15 June 2017.

At Perdigão, 184 sonic anemometers were mounted on 48 meteorological towers, which provided an unprecedented density of instruments in such a limited domain (Figure 1). Observations from the sonic anemometers (a mix of Campbell Scientific CSAT3, METEK uSonic, Gill WindMaster, and YOUNG Model 81000 instruments) were recorded at a 20-Hz frequency. The height of the towers ranged from 2 m to 100 m, with the sonic anemometers mounted at various levels on each tower, as detailed in Table 1 and summarized in the histogram in Figure 2, allowing for an extensive survey of the variability of the wind flow in

**Table 1.** Heights where sonic anemometers were mounted on the meteorological towers at the Perdigão field campaign.

| Tower height | Sonic anemometer heights (m AGL) | Number of towers |
|---|---|---|
| 2 m | 2 | 1 |
| 10 m | 10 | 5 |
| | 2, 10 | 5 |
| 20 m | 10, 20 | 10 |
| | 2, 10, 20 | 6 |
| | 2, 4, 6, 8, 10, 12, 20 | 4 |
| 30 m | 10, 30 | 3 |
| | 2, 4, 6, 8, 10, 12, 20, 30 | 5 |
| 60 m | 10, 20, 30, 40, 60 | 5 |
| | 2, 4, 6, 8, 10, 12, 20, 30, 40, 60 | 1 |
| 100 m | 10, 20, 30, 40, 60, 80, 100 | 3 |
| | Total number of towers | 48 |
| | Total number of sonic anemometers | 184 |

the boundary layer. Data from the sonic anemometers have been tilt-corrected following the planar fit method (Wilczak et al., 2001), and rotated into a geographic coordinate system.

To classify atmospheric stability, we calculate the Obukhov length $L$ from each sonic anemometer as

$$L = -\frac{\overline{\theta_v} \cdot u_*^3}{k \cdot g \cdot \overline{w'\theta_v'}}. \tag{1}$$

$\theta_v$ is the virtual potential temperature ($K$, here approximated as the sonic temperature); $u_*$ is the friction velocity ($\mathrm{m\,s^{-1}}$); $k = 0.4$ is the von Kármán constant; $g = 9.81\,\mathrm{m\,s^{-2}}$ is the gravity acceleration; and $\overline{w'\theta_v'}$ is the kinematic buoyancy flux ($\mathrm{m\,K\,s^{-1}}$). For atmospheric stability, we classify unstable conditions as $\zeta = z/L < $ -0.02; and stable conditions as $\zeta > 0.02$; nearly-neutral conditions as $|\zeta| \leq 0.02$.

## 2.2 TKE dissipation rate from sonic anemometers

TKE dissipation rate from the sonic anemometers on the meteorological towers is calculated from the second-order structure function $D_U(\tau)$ of the horizontal velocity $U$ (Muñoz-Esparza et al., 2018):

$$\epsilon = \frac{1}{U\tau}[aD_U(\tau)]^{3/2} \tag{2}$$

where $\tau$ indicates the time lags over which the structure function is calculated, and $a = 0.52$ is the one-dimensional Kolmogorov constant (Paquin and Pond, 1971; Sreenivasan, 1995). We calculate $\epsilon$ every 30 s, and then average values at a 30-minute resolution. At each calculation of $\epsilon$, we fit experimental data to the Kolmogorov model (Kolmogorov, 1941; Frisch,

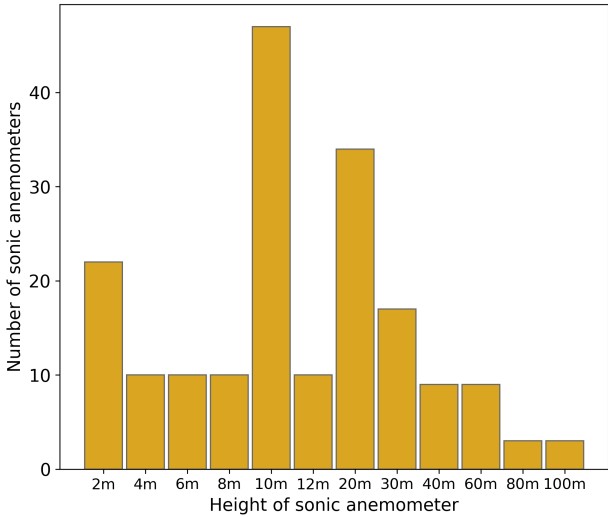

**Figure 2.** Histogram of the heights AGL of the 184 sonic anemometers considered in this analysis.

1995) using time lags between $\tau_1 = 0.1$ s and $\tau_2 = 2$ s, which represent a conservative choice to approximate the inertial subrange (Bodini et al., 2018).

To account for the uncertainty in the calculation of $\epsilon$, we apply the law of combination of errors, which tracks how random errors propagate through a series of calculations (Barlow, 1989). We apply this method to equation 2 and quantify the fractional standard deviation in the $\epsilon$ estimates (Piper, 2001; Wildmann et al., 2019) as

$$\sigma_\epsilon = \frac{3}{2} \frac{\sigma_I}{I} \epsilon \tag{3}$$

where $I$ is the sample mean of $\tau^{-2/3} D_U(\tau)$, and $\sigma_I^2$ is its sample variance. To perform our analysis only on lowly-uncertain $\epsilon$ values, we discard dissipation rates characterized by $\sigma_\epsilon > 0.05$. About 3% of the data are discarded based on this criterion.

As additional quality controls, to exclude tower wake effects, data have been discarded when the recorded wind direction was within $\pm\ 30°$ of the direction of the tower boom. Data during precipitation periods (as recorded by a precipitation sensor on the tower 'riSW06' on the southwest ridge) have also been discarded from further analysis. After all the quality controls have been applied, a total (from all sonic anemometers) of over 284,000 30-minute average $\epsilon$ data remains for the analysis.

## 3 Accuracy of current parameterization of TKE dissipation rate in mesoscale models

Before testing the performance of machine-learning algorithms in predicting TKE dissipation rates, we first assess the current accuracy of the parameterization of $\epsilon$ in numerical models. In the Weather Research and Forecasting model (WRF, Skamarock et al. (2005)), the most widely used numerical weather prediction model, turbulence in the boundary layer can be represented with several planetary-boundary-layer (PBL) schemes, most of which implicitly assume a local balance between turbulence

production and dissipation. Among the different PBL schemes, the MYNN scheme is one of the most commonly chosen. Turbulence dissipation rate in MYNN is given (Nakanish, 2001) as a function of TKE as

$$\epsilon = \frac{(2\,\mathrm{TKE})^{3/2}}{B_1\,L_M} \tag{4}$$

where $B_1 = 24$, and the master length scale, $L_M$, is defined with a diagnostic equation, based on large-eddy simulations, as a function of three other length scales

$$\frac{1}{L_M} = \frac{1}{L_S} + \frac{1}{L_T} + \frac{1}{L_B}. \tag{5}$$

$L_S$ is the length scale in the surface layer, given by

$$L_S = \begin{cases} \kappa\,z/3.7 & \zeta \geq 1 \\ \kappa\,z\,(1 + 2.7\,\zeta)^{-1} & 0 \leq \zeta < 1 \\ \kappa\,z\,(1 - \alpha_4\,\zeta)^{0.2} & \zeta < 0 \end{cases} \tag{6}$$

where $\kappa = 0.4$ is the von Kármán constant, $\zeta = z/L$ (with $L$ the Obukhov length), $\alpha_4 = 100.0$.

$L_T$ is the length scale depending upon the turbulent structure of the PBL (Mellor and Yamada, 1974), defined as

$$L_T = \alpha_1 \frac{\int_0^\infty q\,z\,dz}{\int_0^\infty q\,dz} \tag{7}$$

where $q = \sqrt{2\,TKE}$, and $\alpha_1 = 0.23$.

$L_B$ is a length scale limited by the buoyancy effect, given by

$$L_B = \begin{cases} \alpha_2\,q/N & \partial\Theta/\partial z > 0 \text{ and } \zeta \geq 0 \\ \frac{\alpha_2\,q + \alpha_3\,q\,(q_c/L_T\,N)^{1/2}}{N} & \partial\Theta/\partial z > 0 \text{ and } \zeta < 0 \\ \infty & \partial\Theta/\partial z \leq 0 \end{cases} \tag{8}$$

with $N$ the Brunt-Väisälä frequency, $\Theta$ the mean potential temperature, $\alpha_2 = 1.0$, $\alpha_3 = 5.0$, and $q_c = \left[(g/\Theta_0 \overline{w'\theta'} L_T)\right]^{1/3}$.

From the available observations from the meteorological towers at Perdigão, only $L_S$ can be determined, while the calculation of $L_T$ and $L_B$ would only be possible with critical assumptions about the vertical profile of TKE. Therefore, we decide to approximate $L_M$ as

$$\frac{1}{L_M} \approx \frac{1}{L_S}. \tag{9}$$

By doing so, $L_M$ is overestimated (proof shown in the Supplement), which in turn implies that $\epsilon$ calculated using Eq. (4) will be underestimated.

To evaluate the accuracy of the MYNN parameterization of $\epsilon$, we calculated, using 30-minute average data, the parameterized $\epsilon$ using Eq. (4) (with the approximation in Eq. (9)) from all of the 184 sonic anemometers considered in the study, and

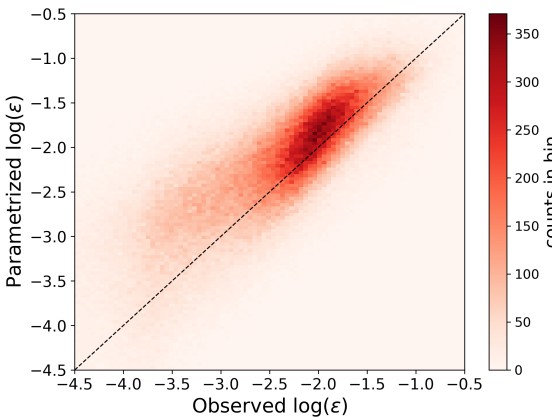

**Figure 3.** Density histogram showing the comparison between observed and MYNN-parameterized $\epsilon$ from the 184 sonic anemometers at Perdigão.

compared with the observed values of TKE dissipation rate (Figure 3) derived from the sonic anemometers with Eq. (2). Given the extremely large range of variability of $\epsilon$, we calculate all the error metrics using the logarithm of predicted and observed $\epsilon$. The TKE dissipation rate predicted by the MYNN parameterization shows, on average, a large positive bias compared to the observed values, with a mean bias of +12% in terms of the logarithm of $\epsilon$, +47% in terms of $\epsilon$. The root-mean-square error (RMSE) is 0.61, and the mean absolute error (MAE) is 0.46. The observed bias would be even larger if $L_M$ was calculated including all the contributions according to Eq. (5), and not $L_s$ only as in our approximation. Therefore, while the approximation in Eq. (9) is major and could be eased by making assumptions on the vertical profile of TKE at Perdigão, it does not affect the conclusion of a high inaccuracy in the MYNN parameterization of $\epsilon$.

Different atmospheric stability conditions give different biases. Figure 4 compares observed and parameterized $\epsilon$ values for stable and unstable conditions, classified based on $\zeta = z/L$, measured at each sonic anemometer, according to the thresholds described in Section 2.1. Stable cases show the largest bias (mean of +24% in terms of the logarithm of $\epsilon$, +101% in terms of $\epsilon$), whereas for unstable conditions the bias is smaller (mean of +6% in terms of the logarithm of $\epsilon$, +19% in terms of $\epsilon$). The MYNN parameterization of $\epsilon$ is therefore especially inadequate to represent small values of $\epsilon$, which mainly occur in stable conditions.

Different heights also impact the accuracy of the parameterization of $\epsilon$. As shown in Figure 5, the mean bias in parameterized $\log(\epsilon)$ decreases with height, while its spread (quantified in terms of the standard deviation of the bias at each height) does not show a large variability at different levels. Close to the surface (data from the sonic anemometers at 2 m AGL), a mean bias (in logarithmic space) of about +25% is found, whereas for the sonic anemometers at 100 m AGL, we find a mean bias of just $\sim$ +3%. This difference in bias with height becomes much larger if the bias is calculated on the actual $\epsilon$ values (and not on their logarithm). We obtain comparable results when computing the bias in the MYNN parameterization only for the sonic

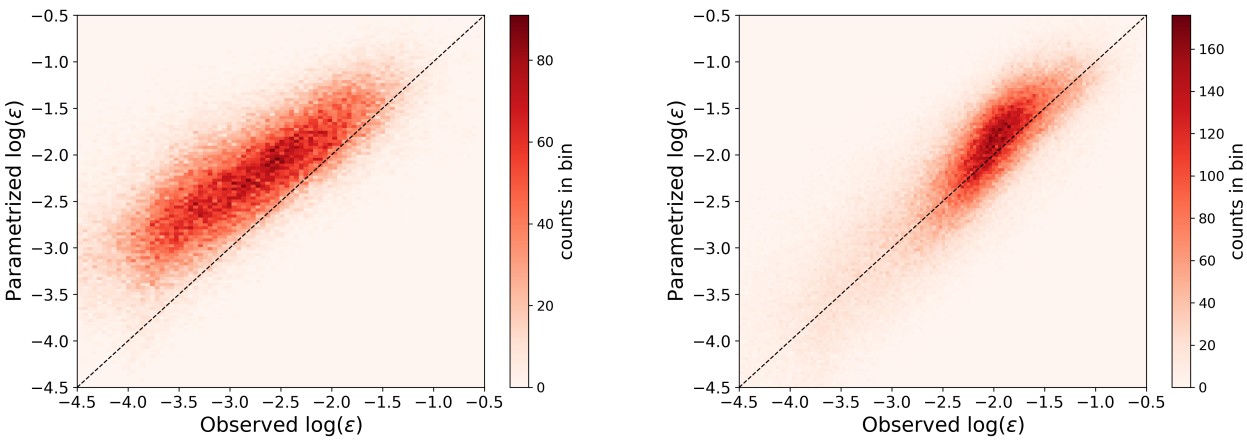

**Figure 4.** Density histogram showing the comparison between observed and MYNN-parameterized $\epsilon$ from the 184 sonic anemometers at Perdigão for stable conditions (left) and unstable conditions (right), as quantified by $\zeta = z/L$ calculated at each sonic anemometer.

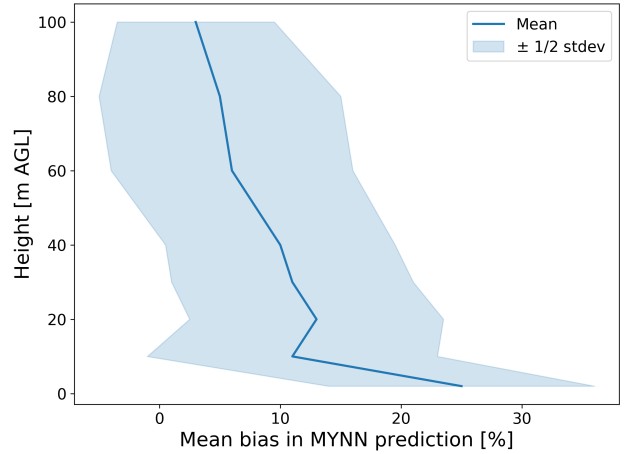

**Figure 5.** Bias in the MYNN-parameterized $\log(\epsilon)$ at different heights, as calculated from the 184 sonic anemometers at Perdigão.

anemometers mounted on the three 100-m meteorological towers (Figure shown in the Supplement), thus confirming that the observed trend is not due to the larger variability of the conditions sampled by the more numerous sonics at lower heights. Therefore, our results show how the MYNN formulation fails in accurately representing atmospheric turbulence especially in the lowest part of the boundary layer.

## 4 Machine-learning algorithms

To test the power of machine learning to improve the numerical representation of the TKE dissipation rate, we consider three learning algorithms in this study: multivariate linear regression, multivariate third-order polynomial regression, and random forest. Given the proof-of-concept nature of this analysis in proving the capabilities of machine learning to improve numerical model parameterizations, we defer an exhaustive comparison of different machine-learning models to a future study, and only consider relatively simple algorithms in the present work. The learning algorithms are trained and tested to predict the logarithm of $\epsilon$ using 30-minute average data. For all but the random forest algorithm, the data were scaled and normalized. No data imputation was performed, and missing data were removed from the analysis.

For the purpose of machine-learning algorithms, the data set has to be divided into three subsets: training, validation, and testing sets (Friedman et al., 2001). The algorithms are first trained multiple times with different hyperparameters (model parameters whose values are set before the training phase and that control the learning process) on the training set, then the validation set is used to choose the best set of hyperparameters, and finally the predicting performance of the trained algorithm is assessed on the testing set. Usually, the data set is split randomly into training, validation, and testing sets. However, as the data used in this study consist of observations averaged every 30 minutes, data in contiguous time stamps are likely characterized by some auto-correlation. Therefore, the traditional random split between training and testing data would lead to an artificially enhanced performance of the machine-learning algorithms, which would be tested on data with a large auto-correlation with the ones used for the training. Therefore, here we use one concurrent week of the data for testing (∼17% of the data), whereas the other 5 weeks are split between training (4 weeks, 66% of the data) and validation (1 week, 17% of the data). The 1-week testing period is shifted continuously throughout the considered 6 weeks of observations at Perdigão, so that each model is trained and its prediction performance tested six times. For each algorithm, we evaluate the overall performance based on the RMSE between the actual and predicted (logarithm of) $\epsilon$, averaged over the different week-long testing periods.

Before testing the models, however, it is important to avoid overfitting by setting the values of hyperparameters. Each learning algorithm has specific model-specific hyperparameters that need to be considered, as will be specified in the description of each algorithm. To test different combinations of hyperparameters and determine the best set, we use cross validation with randomized search, with 20 parameter sets sampled for each learning algorithm. For each set of hyperparameters, the RMSE between the actual and predicted $\log(\epsilon)$ in the validation test is calculated. For each model, we select the hyperparameter combination (among the ones surveyed in the cross validation) that leads to the lowest mean (across the five validation sets) RMSE. We then use this set as the final combination for assessing the performance of the models on the testing set. Overall, the procedure is repeated six times, by shifting the 1-week testing set (Figure 6).

In the following paragraphs, we describe the main characteristics of the three machine-learning algorithms used in our study. A more detailed description can be found in machine-learning textbooks (Hastie et al., 2009; Géron, 2017).

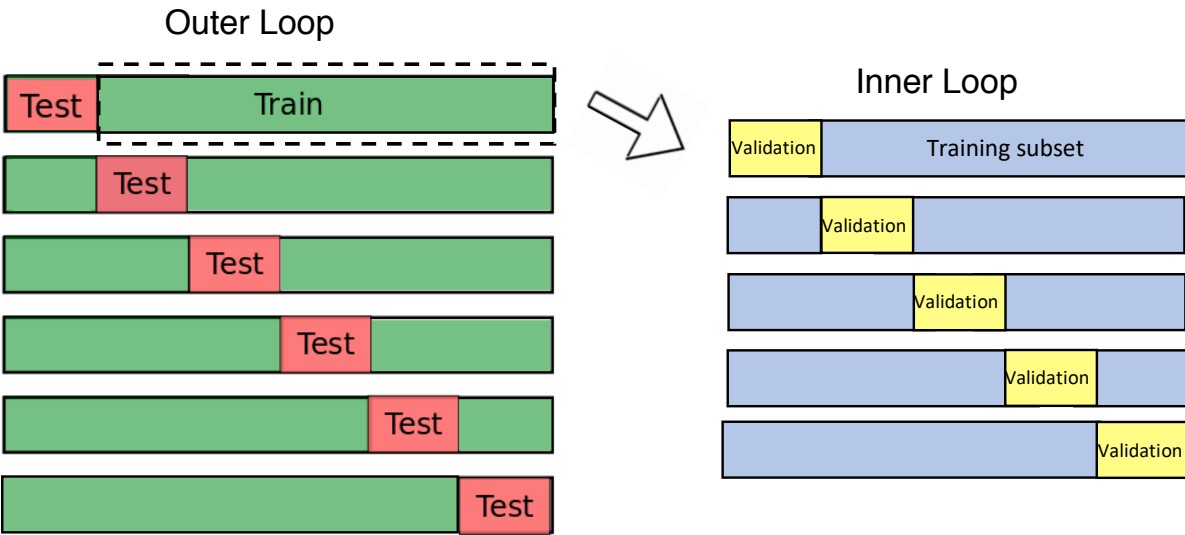

**Figure 6.** Cross-validation approach used to evaluate the performance of the machine-learning models considered in this study.

### 4.1 Multivariate linear regression

To check whether simple learning algorithms can improve the current numerical parameterization of $\epsilon$, we test the accuracy of multivariate linear regression

$$\log(\hat{\epsilon}) = \theta_0 + \theta_1\,x_1 + \theta_2\,x_2 + ... + \theta_n\,x_n \tag{10}$$

where $\hat{\epsilon}$ is the machine-learning predicted value of $\epsilon$, $n$ is the number of features used to predict $\epsilon$ (here 6 - see Section 4.4), $x_i$ is the $i^{\text{th}}$ feature value, and $\theta_j$ is the $j^{\text{th}}$ model weight.

To avoid training a model that overfits the data, regularization techniques need to be implemented, so that the learning model is constrained: the fewer degrees of freedom the model has, the harder it will be for it to overfit the data. We use Ridge regression (Hoerl and Kennard, 1970) (`Ridge` in python's library Scikit-learn) to constrain the multivariate regression. Ridge regression constrains the weights of the model $\theta_j$ to have them stay as small as possible. The Ridge regression is achieved by adding a regularization term to the cost function (MSE)

$$J(\theta) = MSE(\theta) + \alpha \sum_{i=1}^{n} \theta_i^2 \tag{11}$$

where the hyperparameter $\alpha$ controls how much the model will be regularized. The optimal value of the hyperparameter $\alpha$ is determined by cross validation, as described earlier, with values sampled in the range from 0.1–10.

### 4.2 Multivariate third-order polynomial regression

Multivariate polynomial regression can easily be achieved by adding powers of each input feature as new features. The regression algorithm is then trained as a linear model on this extended set of features. For a third-order polynomial regression, the

model becomes

$$\log(\hat{\epsilon}) = \theta_0 + \sum_{i=1}^{n} \theta_i \, x_i + \sum_{i=1}^{n} \theta_{ii} \, x_i^2 + \sum_{i=1}^{n-1} \sum_{j=i+1}^{n} \theta_{ij} \, x_i \, x_j$$

$$+ \sum_{i=1}^{n} \theta_{iii} \, x_i^3 + \sum_{i=1}^{n} \sum_{j \neq i} \theta_{iij} \, x_i^2 \, x_j$$

$$+ \sum_{i=1}^{n-2} \sum_{j=i+1}^{n-1} \sum_{k=j+1}^{n-1} \theta_{ijk} \, x_i \, x_j \, x_k \tag{12}$$

Ridge regression (`Ridge` in Scikit-learn) is used again to constrain the multivariate polynomial regression, with the hyperparameter $\alpha$ in Eq. (11) determined via cross validation, with values sampled in the range from 1–2000.

## 4.3 Random forest

Random forests (`RandomForestRegressor` in Scikit-learn) combine multiple decision trees to provide an ensemble prediction. A decision tree can learn patterns and then predict values by recursively splitting the training data based on thresholds of the different input features. As a result, the data are divided into groups, each associated with a single predicted value of $\epsilon$, calculated as the average target value (of the observed $\epsilon$) of the instances in that group.

As an ensemble of decision trees, a random forest trains them on different random subsets of the training set. Once all the predictors are trained, the ensemble (i.e., the random forest) can make a prediction for a new instance by taking the average of all the predictions from the single trees. In addition, random forests introduce some extra randomness when growing trees: instead of looking for the feature that, when split, reduces the overall error the most when splitting a node, a random forest searches for the best feature among a random subset of features.

Decision trees make very few assumptions about the training data. As such, if unconstrained, they will adapt their structure to the training data, fitting them closely, and most likely overfitting them, without then being able to provide accurate predictions on new data. To avoid overfitting, regularization can be achieved by setting various hyperparameters that insert limits to the structure of the trees used to create the random forests. Table 2 describes which hyperparameters we considered for the random forest algorithm. For each hyperparameters listed, we include the range of values that are randomly sampled in the cross-validation search to form the twenty sets of hyperparameters considered in the training phase.

## 4.4 Input features for machine-learning algorithms

Given the large variability of $\epsilon$, which can span several orders of magnitude (Bodini et al., 2019b), we apply the machine-learning algorithms to predict the *logarithm* of $\epsilon$. To select the set of input features used by the learning models, we take advantage of the main findings of the observational studies on the variability of $\epsilon$ to select as inputs both atmospheric- and terrain-related variables to capture the impact of topography on atmospheric turbulence. For each variable, we calculate and use in the machine learning algorithms 30-minute average data, to reduce the high autocorrelation in the data and limit the

**Table 2.** Hyperparameters considered for the random forest algorithm.

| Hyperparameter | Meaning | Sampled values |
|---|---|---|
| Number of estimators | Number of trees in the forest | 10 - 250 |
| Maximum depth | Maximum depth of the tree | 1 - 50 |
| Maximum number of leaf nodes | Maximum number of leaf nodes in the decision tree | 2000 - 500,000 |
| Maximum number of features | Number of features to consider when looking for the best split | 1 - 6 |
| Minimum number of samples to split | Minimum number of samples required to split an internal node | 1 - 200 |
| Minimum number of samples for a leaf | Minimum number of samples required to be at a leaf node | 1 - 50 |

impact of the high-frequency large variability of turbulent quantities. We use the following input features (calculated at the same location and height as $\epsilon$) for the three learning algorithms considered in our study:

- wind speed (WS), which has been shown to have a moderate correlation with $\epsilon$ (Bodini et al., 2018);

- the logarithm of TKE, which is expected to have a strong connection with $\epsilon$ according to Eq. (4), calculated as

$$\log(\text{TKE}) = \log\left[\frac{1}{2}\left(\sigma_u^2 + \sigma_v^2 + \sigma_w^2\right)\right] \tag{13}$$

where the variances of the wind components are calculated over 30-minute intervals. The choice of using the *logarithm* of TKE is justified by the fact Eq. 4 suggests this quantity is linearly related to the logarithm of $\epsilon$;

- the logarithm of friction velocity $u_*$, which is calculated as

$$u_* = (\overline{u'w'}^2 + \overline{v'w'}^2)^{1/4}. \tag{14}$$

An averaging period of 30 minutes (De Franceschi and Zardi, 2003; Babić et al., 2012) has been used to apply the
235 Reynolds decomposition and calculate average quantities and fluctuations.

- the log-modulus transformation (John and Draper, 1980) of the ratio $\zeta = z_\text{son}/L$, where $z_\text{son}$ is the height above the ground of each sonic anemometer, and $L$ is the 30-minute average Obukhov length:

$$\text{sign}(\zeta)\log(|\zeta| + 1) \tag{15}$$

The use of $\zeta$ is justified within the context of the Monin Obukhov similarity theory (Monin and Obukhov, 1954). The
240 use of the logarithm of $\zeta$ is consistent with the use of the logarithm of $\epsilon$ as target variable. Finally, the log-modulus transformation allows for the logarithm to be calculated on negative values of $\zeta$ and be continuous in zero.

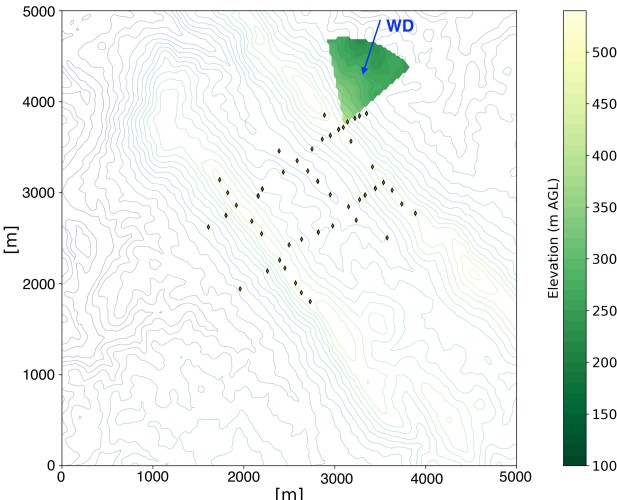

**Figure 7.** Example of an upwind terrain elevation sector with a 1-km radius centered on the location of one of the meteorological towers at Perdigão.

- the standard deviation $\text{std}(z_{\text{terr}})$ of the terrain elevation in a 1-km radius sector centered on the measurement point (i.e., the location of the sonic anemometer). The angular extension of the sector is set equal to $\pm 30°$ from the recorded 30-minute average wind direction (an example is shown in Figure 7). While we acknowledge that some degree of arbitrariness lies
in the choice of this variable to quantify the terrain influence, it represents a quantity that can easily be derived from numerical models, should our approach be implemented for practical applications, to capture the influence of upwind topography to trigger turbulence. To compute this variable, we use Shuttle Radar Topography Mission (SRTM) 1 Arc-Second Global data, at 30 m horizontal resolution.

- the mean vegetation height $\overline{h_{\text{veg}}}$ in the upwind 1-km radius sector centered on the measurement point. Given the forested
nature of the Perdigão region, we expect canopy to have an effect in triggering turbulence, especially at lower heights. To compute this variable, we use data from a lidar survey during the season of the field campaign, at a 20 m horizontal resolution.

The distribution of the input features and of $\log(\epsilon)$ are shown in the Supplement.

While we acknowledge that the input features are not fully uncorrelated, we found that including all these features provides a
255 better predictive power for the learning algorithms, despite negatively affecting the computational requirements of the training phase. The application of principal component analysis can help reduce the number of dimensions in the input features while preserving the predictive power of each, but it is beyond the scope of the current work.

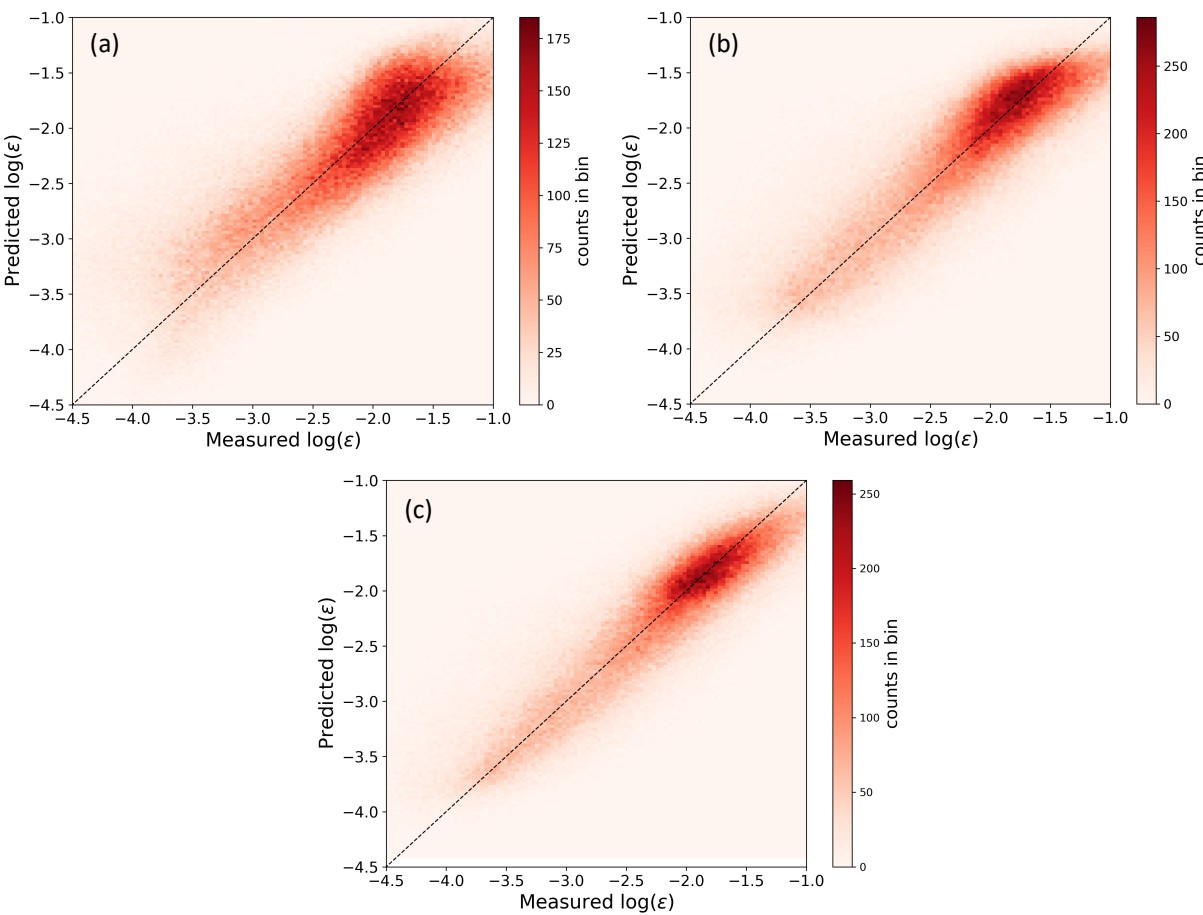

**Figure 8.** Density histogram showing the comparison, performed on the testing set, between observed and machine-learning-predicted $\epsilon$ from a multivariate linear regression (a), a multivariate third-order polynomial regression (b), and a random forest (c).

## 5   Results

### 5.1   Performance of machine-learning algorithms

To evaluate the prediction performance of the three machine-learning algorithms we considered, we use, for each method, a density histogram showing the comparison between observed and machine-learning-predicted $\epsilon$ (Figure 8). The prediction from all the considered learning algorithms do not show a significant mean bias, as found in the MYNN representation of $\epsilon$. As specific error metrics, we compare RMSE and MAE of the machine-learning predictions with what we obtained from the MYNN parameterization, with the caveat that while the MYNN scheme is thought to provide a universal representation of $\epsilon$, the

machine-learning models have been specifically trained on data from a single field campaign. Each machine-learning algorithm was tested on six 1-week-long testing periods, as described in Section 4. For each method we present the RMSE and MAE

averaged across the different testing periods. Even the simple multivariate linear regression (Figure 8-a) improves, on average, on MYNN. Overall, the average RMSE (0.47) is 23% smaller than the MYNN parameterization, and the average MAE (0.36) is 22% lower than the MYNN prediction. The multivariate third-order polynomial regression provides an additional improvement

(Figure 8-b) for the representation of $\epsilon$, with the average RMSE (0.44) over 28% smaller than the MYNN parameterization, and the average MAE (0.33) 28% lower than the MYNN representation. The additional input features created by the polynomial model allow for an accurate prediction of $\epsilon$ even at the low turbulence regime. Finally, the random forest further reduces the spread in machine-learning predicted $\epsilon$, with the RMSE (0.40) reduced by about 35% from the MYNN case, and the MAE (0.29) by 37%, with no average bias between observed and predicted values of $\epsilon$.

Table 3 summarizes the performance of all the considered algorithms. We note that, because the length scale approximation we made in calculating MYNN-predicted $\epsilon$ led to a better agreement with the observed values compared to what would be obtained using the full MYNN parameterization, the RMSE and MAE for the MYNN case would in reality be higher than what we report here, and so the error reductions achieved with the machine-learning algorithms would even be greater than the numbers shown in the Table.

**Table 3.** Performance of the machine-learning algorithms trained and tested at Perdigão, measured in terms of RMSE and MAE between the logarithm of observed and MYYN-parameterized $\epsilon$.

|  | MYNN parameterization | Linear regression | Third-order polynomial regression | Random forest |
|---|---|---|---|---|
| RMSE | 0.61 | 0.47 | 0.44 | 0.40 |
| % change in RMSE |  | -23% | -28% | -35% |
| MAE | 0.46 | 0.36 | 0.33 | 0.29 |
| % change in MAE |  | -22% | -28% | -37% |

Given the large gap in the performance of the MYNN parameterization of $\epsilon$ between stable and unstable conditions, it is worth exploring how the machine learning algorithms perform in different stability conditions. To do so, we train and test two separate random forests: one using data observed in stable conditions, the other one for unstable cases. We find that both algorithms eliminate the bias observed in the MYNN scheme (Figure 9). The random forest for unstable conditions provides, on average, more accurate predictions (RMSE = 0.37, MAE = 0.28) compared to the algorithm used for stable cases (RMSE

0.44, MAE = 0.33), thus confirming the complexity in modeling atmospheric turbulence in quiescent conditions. However, when the error metrics are compared to those of the MYNN parameterization, the random forest for stable conditions provides the largest relative improvement, with a 50% reduction in MAE, while for unstable conditions the reduction is of 20%.

## 5.2 Physical interpretation of machine-learning results

Not only machine learning techniques provide accuracy improvements to represent atmospheric turbulence, but additional
290 insights on the physical interpretation of the results can - and should - be achieved. In particular, random forests allow for an

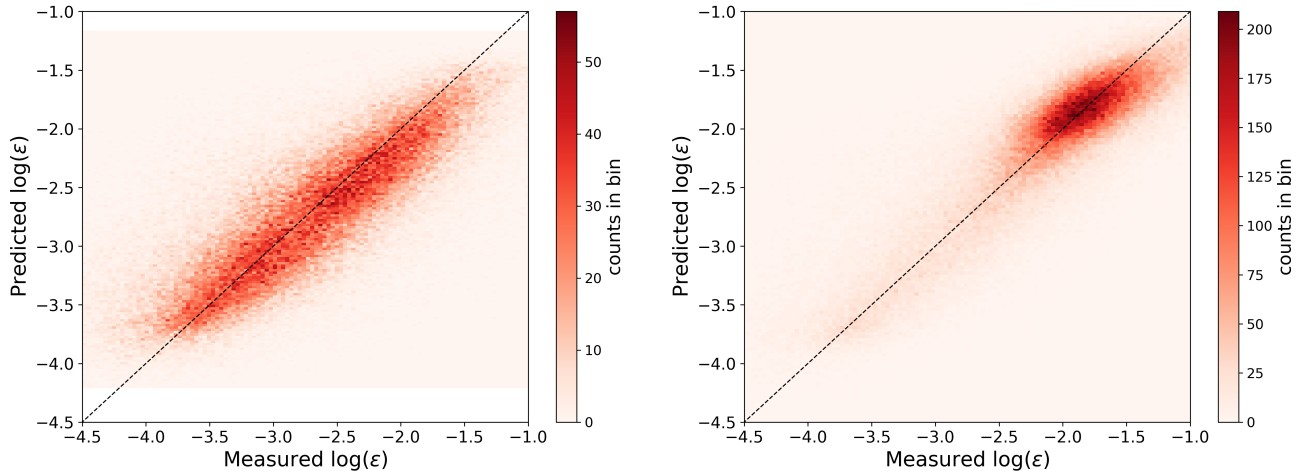

**Figure 9.** Density histogram showing the comparison, performed on the testing set, between observed and machine-learning-predicted $\epsilon$ from a random forest for stable conditions (left) and unstable conditions (right).

**Table 4.** Feature importance classification as derived from the random forest.

| Input feature | Feature importance |
|:---:|:---:|
| $\log(\text{TKE})$ | 47% |
| $\log(u_*)$ | 24% |
| $\text{sign}(\zeta)\log(|\zeta|+1)$ | 13% |
| WS | 11% |
| $\text{std}(z_{terr})$ | 3% |
| $\overline{h_{veg}}$ | 2% |

assessment of the relative importance of the input features used to predict (the logarithm of) $\epsilon$. The importance of a feature is calculated by looking at how much the tree nodes that use that feature reduce the MSE on average (across all trees in the forest), weighted by the number of times the feature is selected. Table 4 shows the feature importance for the six input features we used in this study. The feature importance results are affected by the correlation between some of the input features used in the models. We find how the logarithm of turbulence kinetic energy is the preferred feature for tree splitting, with the largest importance (47%) in reducing the prediction error for $\epsilon$ in the random forest. This result, which can be expected as both TKE and $\epsilon$ are variables connected to turbulence in the boundary layer, agrees well with the current formulation of the MYNN parameterization of $\epsilon$, which includes TKE as main term. As TKE is correlated to $u_*$ and $L$, we find that the decision trees more often split the data based on TKE, so that the feature importance of its correlated variables is found to be lower. The limitations of the Monin-Obukhov similarity theory (Monin and Obukhov, 1954) in complex terrain might also be an additional cause for the relatively low feature importance of the feature associated with $L$. The standard deviation of the upwind elevation and

the mean vegetation height have the lowest importance, of respectively 3% and 2%. Though not negligible, the importance of topography and canopy might increase by considering different parameters that could better encapsulate their effect. We have tested how the feature importance varies when considering several random forests, each trained and tested with data from all the sonic anemometers at a single height only, and did not find any significant variation of the importance of the considered variables in predicting $\epsilon$ (plot shown in the Supplement).

Finally, to assess the dependence of TKE dissipation rate on the individual features considered in this study, Figure 10 shows partial dependence plots for the input features considered in the analysis. These are obtained, for each input feature, by applying the machine-learning algorithm (here, random forests) multiple times with the other feature variables constant (at their means) while varying the target input feature and measuring the effect on the response variable (here, $\log(\epsilon)$). In each plot, the values on the y-axes have not been normalized, so that large ranges show a strong dependence of $\log(\epsilon)$ on the feature, whereas small ranges indicate weaker dependence. The strong relationship between $\epsilon$ and TKE is confirmed, as the range shown on its y-axis is the largest among all features. As TKE increases, so does $\epsilon$. A similar trend, tough with a weaker influence, emerges when considering the dependence of $\epsilon$ on friction velocity. The relationship between $\epsilon$ and wind speed shows a less clear trend, and with a weaker dependence: $\epsilon$ increases for 30-minute average wind speeds up to $\sim$2 m s$^{-1}$, and then decreases for stronger wind speed values. A more distinct trend could emerge when considering data averaged at shorter time periods. The dependence between TKE dissipation and atmospheric stability shows a moderate impact, with stable conditions (positive values of the considered metric) showing smaller $\epsilon$ values compared to unstable cases (negative values of the considered metric). Interestingly, the largest $\epsilon$ values seems to be connected to neutral cases. Finally, both terrain elevation and vegetation height show weak impact on determining the values of $\epsilon$, as quantified by the narrow range of values sampled on the y-axis for these two variables.

## 6 Conclusions

Despite turbulence being a fundamental quantity for the development of multiple phenomena in the atmospheric boundary layer, the current representations of TKE dissipation rate ($\epsilon$) in numerical weather prediction models suffer from large inaccuracies. In this study, we quantified the error introduced in the MYNN parameterization of $\epsilon$ by comparing predicted and observed values of $\epsilon$ from 184 sonic anemometers from 6 weeks of observations at the Perdigão field campaign. A large positive bias (average +12% in logarithmic space, +47% in natural space) emerges, with larger errors found in atmospheric stable conditions. The need for a more accurate representation of $\epsilon$ is therefore clearly demonstrated.

The results of this study show how machine learning can provide new ways to successfully represent TKE dissipation rate from a set of atmospheric and topographic parameters. Even ultrasimple models such as a multivariate linear regression can provide an improved representation of $\epsilon$ compared to the current MYNN parameterization. More sophisticated algorithms, such as a random forest approach, lead to the largest benefits, with over a 35% reduction in the average error introduced in the parameterization of $\epsilon$, and eliminate the large bias found in it, for the Perdigão field campaign. When considering stable conditions only, the reduction in average error reaches 50%. Although the generalization gap between the universal nature of

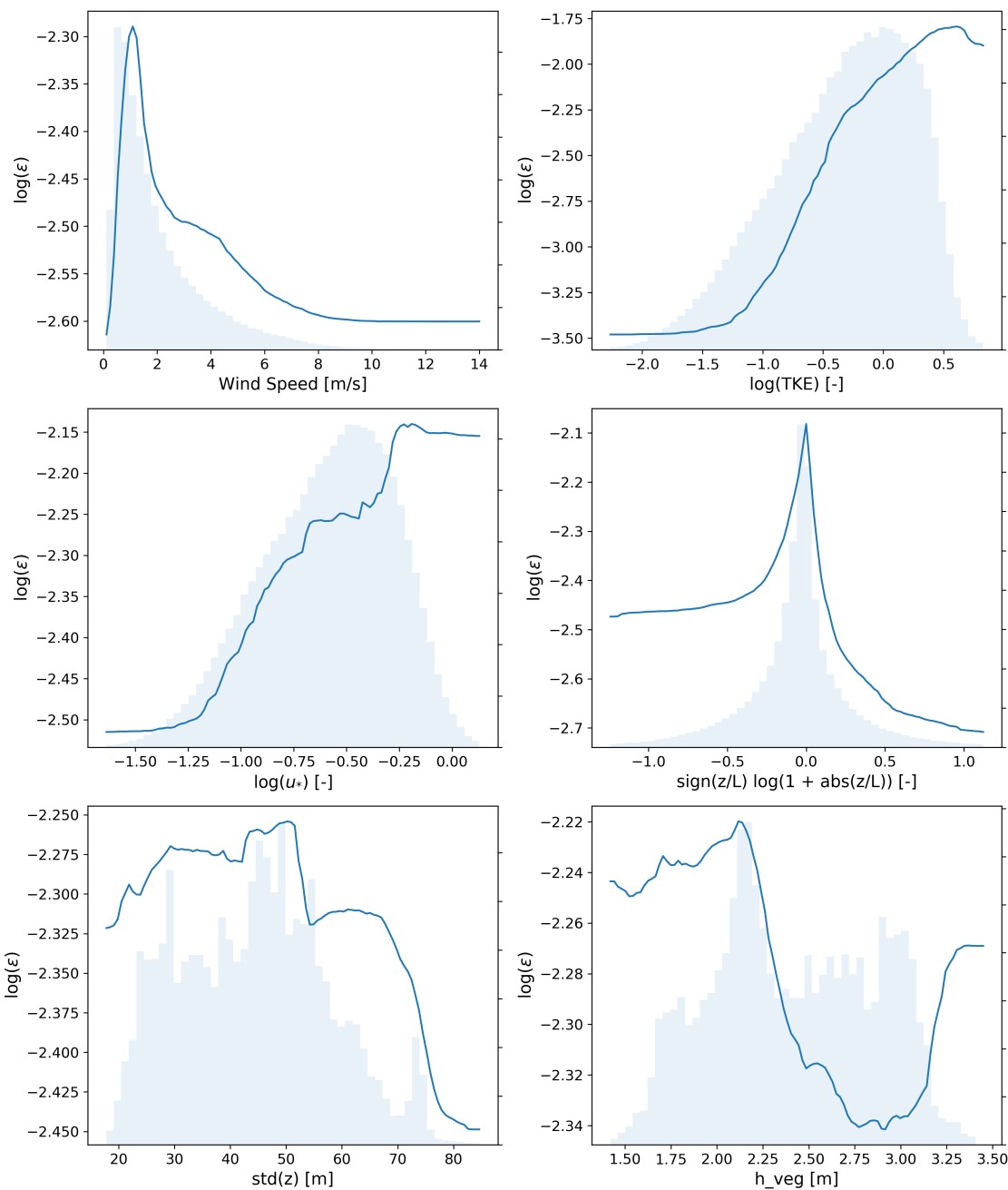

**Figure 10.** Partial dependence plots for the input features used in the analysis. Distributions of the considered features are shown in the background.

the MYNN parameterization of $\epsilon$ and the campaign-specific training and testing of the machine-learning models needs to be acknowledged, the results of this study can be considered as a proof of concept of the potentialities of machine-learning-based representations of complex atmospheric processes.

Multiple opportunities exist to extend the work presented here. In the future, additional learning algorithms, such as support vector machines and extremely randomized trees, should be considered. Deep learning methods, such as recurrent neural networks, and specifically long-short term memory, which are well-suited for time-series-based problems, could also be considered to obtain a more complete overview of the capabilities of machine-learning techniques for improving numerical representations of $\epsilon$. Moreover, additional input features could be added to the learning algorithms to possibly identify additional variables with a large impact on atmospheric turbulence. Finally, the learning algorithms developed here would need to be tested using data from different field experiments, to understand whether the results obtained in this study can be generalized everywhere. Once the performance of a machine-learning representation of $\epsilon$ has been accurately tested, its implementation in numerical weather prediction models, such as the Weather Research and Forecasting model, should be achieved.

*Code and data availability.* High-resolution data from sonic anemometers on the meteorological towers (UCAR/NCAR, 2019) are available through the EOL project at https://doi.org/10.26023/8X1N-TCT4-P50X. Digital Elevation Model data are taken from the SRTM 1 Arc-Second Global at https://doi.org/10.5066/F7PR7TFT. The vegetation height data are available upon request to Prof. Jose Laginha Palma at the University of Porto. The machine learning code used for the analysis is stored at https://doi.org/10.5281/zenodo.3754710.

*Author contributions.* NB and JKL designed the analysis. NB analyzed the data from the sonic anemometers and applied the machine-learning figures, in close consultation with JKL and MO. NB wrote the paper, with significant contributions from JKL and MO.

*Competing interests.* The authors declare that they have no conflicts of interest.

*Acknowledgements.* We thank the residents of Alvaiade and Vale do Cobrão for their essential hospitality and support throughout the field campaign. In particular, we are grateful for the human and logistic support Felicity Townsend provided to our research group in the field. We thank Prof. Jose Laginha Palma for providing the vegetation data used in the analysis. We thank Dr. Ivana Stiperski for her exceptionally thoughtful review of our discussion paper, which greatly improved the quality of this work. Support to NB and JKL is provided by the National Science Foundation, under the CAREER program AGS-1554055 and the award AGS-1565498. This work utilized the RMACC Summit supercomputer, which is supported by the National Science Foundation (awards ACI-1532235 and ACI-1532236), the University of Colorado Boulder, and Colorado State University. The Summit supercomputer is a joint effort of the University of Colorado Boulder and Colorado State University.

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
