# Peer review of "Can machine learning improve the model representation of TKE dissipation rate in the boundary layer for complex terrain?"

_Geoscientific Model Development, 2020_

## Referee Comment (RC1) · Ivana Stiperski (Referee) · 10 May 2020

Review of "Can machine learning improve the model representation of TKE dissipation rate in the boundary layer for complex terrain?" by Nicola Bodini, Julie K. Lundquist, and Mike Optis

**General comments**

The paper focuses on the turbulence dissipation rate and whether three machine learning techniques can outperform parametrizations of dissipation rate commonly used in numerical models. For this purpose the authors use the Perdigao dataset with an unprecedented number of 184 sonics on towers ranging up to 100m in height. This paper is both timely and relevant as the turbulence dissipation rate is one of the most important turbulence diagnostics and its incorrect representation in models and related biases have wide ranging consequences. The machine learning approach is also the appropriate methodology to tease out the information about possible many influences from such a large dataset and the results are encouraging. Despite its merits, however, the paper still needs to address a number of points listed below, some of which might change the results, before I can recommend it for publication. Given my expertise I focus more on the physical aspect of the paper than details of machine learning. I therefore recommend major revision.

**Specific comments:**

1. **Information on data analysis**

   I find the information on the data post-processing and analysis insufficient.

   - Particularly missing is the information on the averaging times which is confusing. It is stated that the dissipation rates were calculated from the second order structure functions at 30 s intervals, but that TKE was calculated at 2 min intervals (ln 91). Are the other averaging times 30 min (ln 96)? Why is there a difference between the averaging times of different variables and how are they then reconciled for the purposes of machine learning where predictor and response variables need the same length?

   - What is the motivation of calculating TKE at 2 min intervals and not 30 min like the other variables? Are the authors trying to say that the relevant TKE for the dissipation is not the one of the energy containing eddies but the one at smaller scales? Is then $TKE^{2/3}$ calculated at 30s, 30 min intervals or 2 min? And is there other motivation for having TKE and $TKE^{2/3}$ apart for testing for its nonlinear influence

   - Turbulence data (dissipation rates included) calculated at 30s intervals have a large random error due to under-sampling. Are the authors then averaging the 30s dissipation rates and 2 min TKE values to the 30 min period (Ln 96) to reduce this random error?

   - Apart from tilt correction, are data rotated into the mean wind?

   - Given the forested nature of Perdigao, has the displacement height been taken into account? Is it assured that the measurements are above the canopy layer and roughness sublayer or are the authors testing the parametrization irrespective of the PBL layer that is probed?

   - What is the number of data points used when all the quality criteria are satisfied?

2. **Dissipation calculation**

   Given the very large array of different sonic anemometers, can the author discuss if there were any noticeable differences in the estimated dissipation rates? Is aliasing observed at tau = 0.1s for any of the datasets especially in stable conditions? Have

the authors performed a quality control of the dissipation rate based on if the slope of the structure function is really 2/3 (plus/minus some uncertainty interval)?

3. Multivariate linear regression
   - The multivariate linear regression shows the worst results of the machine learning methods used. At a first glance this comes as no surprise given that the dissipation rate is not necessarily related to other variables in a linear way, despite the fact that it is commonly accepted that dissipation and TKE are strongly coupled. The authors also mention that it is due to dissipation rate spanning multiple order of magnitude more than the TKE. However, the method might be underperforming because of a different reason. Since the response variable is the logarithm of the dissipation rate, there is no reason to expect that the predictor variables should be variables themselves rather than logarithms. For example, equation (5) shows that the parametrization of dissipation is related to TKE through:

   $$\epsilon = \frac{TKE^{\frac{2}{3}}}{BL_M}$$

   If we now want to see how logarithm of dissipation rate is related to TKE we see that it is related to the logarithm of TKE and not to TKE itself:

   $$\log_{10} \epsilon \approx \frac{2}{3}\log_{10} TKE - \log_{10} L_M$$

   Indeed, plotting $\log_{10} \epsilon$ vs TKE produces a similar shape to the one observed in Fig. 8, while $\log_{10} \epsilon$ vs $\log_{10} TKE$ are linearly related.
   I expect that the multi-linear regression will produce a much more significant results with better $R^2$ and less bias if the predictor variables (TKE, u*, z/L) are switched with their logarithms. In the logarithmic representation there will also be only one TKE representation necessary. I suspect the same approach will produce an even better result for random forests.
   - I miss the information on what variables were chosen by the multivariate model? The results are only presented for the random forest. With so many related variables the full model should be penalized.
   - Can the authors discuss more in depth their motivation for choosing the parameters they did maybe within the Monin-Obukhov framework or HOST framework for stable conditions.

4. Influence of measurement height
   Given that there are only a couple of towers that are 100 m high I am wondering about the representativity of these very high measurements as they will occupy only a very small fraction of the training. If one uses z/L then this influence will be normalized and will no longer be an issue, however, the authors use height of the sonic $z_{son}$ which is not normalized and therefore subject to representativity issues. In the same way I wonder about the results of Figure 6 in which the mean bias according to height of MYNN is shown. The results for lower heights will include a more varied set of conditions than for higher heights. I would find it justified to compare the bias for different heights only on the towers with similar heights (for example the two 100m towers).

5. Terrain influence
   The terrain influence in the paper is quantified through a standard deviation of the terrain within 1 km upstream of the measurement. I presume that this is because such a variable is readily available in numerical models, but this motivation is missing

in the paper. On the other hand from a physical point of view I am wondering how this variable can be justified. Given the variety of measurement heights in the dataset the flux footprint and therefore also the terrain that influences the measurement is going to vary substantially. It would therefore be good to either motivate the choice in detail, or to present some footprint analysis which convinces us that the choice of 1 km is meaningful.

I also miss the information on how this standard deviation is computed. What is the resolution of the digital elevation model used for this computation? And what is the reasoning behind using standard deviation as opposed to for example slope angle? Given the change of the footprint with height, wouldn't it be more appropriate to estimate the effect of terrain only for measurements with similar height?

6. **Separation according to stability**

Results of Fig 5 show very large difference in the success of the parametrization for stable and unstable stratification. Looking at the results I would say that there is visually almost no need for improvements on the unstable side. With this in mind, I wonder why the approach is then followed which lumps all the data together.

7. **Paper structure**

The paper structure could be improved if machine learning algorithms were introduced before the predictor variables that are used to feed these algorithms.

**Minor points:**

**Ln 35:** "; for example" should be ": for example"

**Figure 1:** It would be good to color the points according to the height of the tower their represent

**Ln 69:** "are recorded" should be "were recorded"

**Figure 2:** given that this figure is only for presentation purposes I would suggest replacing a histogram for a bar plot which correctly represents the measurement heights. This could still be done in some meaningful increments but would not bundle 2m heights under 0 and would not have gaps for say 90 m height which does not exist

**Ln 75-77:** it is not necessary to mention that it is a structure function of horizontal velocity twice in this sentence

**Ln 79:** the part of the sentence "is done using the temporal separation between" is not very clear. Do you mean that you calculate the dissipation rate for lags between 0.1 and 2s by assuming that this is the inertial subrange?

**Ln 87:** algorithms haven't been introduced yet

**Ln 102:** this is not sensible heat flux but buoyancy flux, given that the authors mention no Schotanus correction. Also, is $\theta_v$ really virtual temperature or rather sonic temperature?

**Ln 103:** Why are the authors using values of L to define stability ranges, when it is more common to define them through z/L, where neutral stratification has a clear meaning, whereas L is not as clearly specified?

**Ln 129:** Given the many profiles that exist in the data, I wonder why it is impossible to estimate the $L_T$ and $L_B$ scales. The TKE is not expected to vary so erratically to not be possible to estimate its vertical variability with an analytical function.

**Ln 140:** so is TKE then calculated at 30s?

**Ln 163:** What do you mean by "time stamps with missing data"? Do you mean that only those periods when all the instruments had all the values were used?

**Ln 165 – 166:** What do you mean by hyperparameters? Are you referring to the ones defined in Table 1? This should be referenced here.

**Ln 194:** Mention that Scikit-learn is a python library.

**Ln 195:** what are the variables chosen by the ridge regression?

**Ln 223:** I find this sentence not very clear. Values of what were sampled in the cross-validation search? And what do you mean by five sets of parameters?

**Table 1**: How were these values chosen?

**Ln 232:** How do you explain this "optimistic result" that using a reduced parametrization is actually beneficial to using the full one?

**Ln 252:** Is $R^2$ the adjusted one that takes into account the penalization for overfitting? Are all the variables statistically significant and at which p value?

**Ln 265:** Within Monin-Obukhov similarity theory L is not the relevant variable but z/L. The use of logarithm of (z/L) might improve the importance of this variable.

---

## Referee Comment (RC2) · Anonymous Referee #2 · 24 May 2020

The manuscript "Can machine learning improve the model representation of TKE dissipation rate in the boundary layer for complex terrain?" by Bodini et al. provides an interesting look at using machine learning techniques to generate estimates of TKE dissipation rate and comparing those results to the approach used in the MYNN parameterization. This work should be of interest to the community and provides a useful road map for scientists wanting to apply a similar approach to other data sets. Overall, I think the manuscript will be acceptable for publication in Geoscientific Model Development after relatively minor revisions. The text is generally clearly written and straight forewarned to follow. I wonder, given that focus on data analysis rather than atmospheric model development, if the manuscript is a better fit for Atmosphere Chemistry

and Physics or Atmospheric Measurement Techniques. I leave that, however, up to the editor.

General comments: • Machine learning techniques generally do not increase our physical understanding. The authors try to address this in Section 5.1 and 5.2 where additional analysis is provided. Section 5.2, however, is very brief and should be developed more to provide additional insight into the results. • In section 3, the authors show that the MYNN approach does a reasonable job in unstable conditions, but much worse when the boundary layer is statically stable. I was surprised that the authors didn't carry this analysis into the subsequent sections. It would seem natural to examine the model behavior with stability in Section 5.

Specific comments: 1. Figure 1. I appreciate the histogram shown in Figure 2, but could you also differentiate the points in Figure 1 to indicate measurement heights? Maybe that doesn't work well if the measurements made at a single location are at several heights? 2. Section 2.1: Can you say anything more about how the sonics are distributed on the towers? For example, how many were deployed on the 100 m tower? 3. Lines 78-80: Double check this sentence, the wording seems odd. 4. Line 101: Is the mean potential temperature computed from the sonic data or does it come from a different source? 5. Lines 104-109: Can you point the reader to the terrain data set that was used? What was the resolution of that data set? Does that have any impact on the results? 6. Line 138: I agree that the length scale assumption is the best you can do given the data set that you have, but I think some additional discussion is warranted to help defend that selection. Can you argue that Ls is likely dominate near the surface? 7. Figure 6: You show the mean bias in Figure 6, could bars be added to indicate the standard deviation of the bias? This would help show how significant the biases are. In addition, the figures shows a decrease with height. Is this significant, or could it (at least partially) be related to the horizontal distribution of the measurements taken at different heights? 8. Section 4: It would be helpful if you could include a brief discussion of why you selected these particular algorithms for this application. 9.

Section 5.2: Is there a better header for this section to help the reader understand the importance of the analysis that is presented? 10. Section 5.2: This section seems to end abruptly. Can you guide the reader to anything important? What additional insight is gained from the analysis? What does it tell us about what is controlling the dissipation rate at large values of wind speed and/or TKE?
* * *

---

## Author Response (AR1)

*In this document, the reviewer's comments are in black, the authors' responses are in red.*

The authors thank the reviewer for their thoughtful and productive comments.

**General comments**

The paper focuses on the turbulence dissipation rate and whether three machine learning techniques can outperform parametrizations of dissipation rate commonly used in numerical models. For this purpose, the authors use the Perdigão dataset with an unprecedented number of 184 sonics on towers ranging up to 100m in height. This paper is both timely and relevant as the turbulence dissipation rate is one of the most important turbulence diagnostics and its incorrect representation in models and related biases have wide ranging consequences. The machine learning approach is also the appropriate methodology to tease out the information about possible many influences from such a large dataset and the results are encouraging. Despite its merits, however, the paper still needs to address a number of points listed below, some of which might change the results, before I can recommend it for publication. Given my expertise I focus more on the physical aspect of the paper than details of machine learning. I therefore recommend major revision.

**Specific comments**

1. **Information on data analysis**
   I find the information on the data post-processing and analysis insufficient.
   - Particularly missing is the information on the averaging times which is confusing. It is stated that the dissipation rates were calculated from the second order structure functions at 30 s intervals, but that TKE was calculated at 2 min intervals (ln 91). Are the other averaging times 30 min (ln 96)? Why is there a difference between the averaging times of different variables and how are they then reconciled for the purposes of machine learning where predictor and response variables need the same length?
   We have added details on the variables used in our analysis. Moreover, we have now calculated all variables using the same 30-minute average period. This has been clarified in many places throughout the manuscript.

   - What is the motivation of calculating TKE at 2 min intervals and not 30 min like the other variables? Are the authors trying to say that the relevant TKE for the dissipation is not the one of the energy containing eddies but the one at smaller scales? Is then $TKE^{2/3}$ calculated at 30s, 30 min intervals or 2 min? And is there other motivation for having TKE and $TKE^{2/3}$ a part for testing for its nonlinear influence?
   As stated above, we have now calculated TKE using a 30 minute averaging period. We have also removed $TKE^{2/3}$ from the set of input features used in the analysis to reduce the correlation between the variables used.

   - Turbulence data (dissipation rates included) calculated at 30s intervals have a large random error due to under-sampling. Are the authors then averaging the 30s

dissipation rates and 2 min TKE values to the 30 min period (Ln 96) to reduce this random error?

We have now addressed this issue by calculating dissipation rates every 30s, and then averaging data at a 30-minute resolution. This has been clarified in the manuscript: "We calculate ε every 30 s, and then average values at a 30-minute resolution." And again: "For each variable, we calculate and use in the machine learning algorithms 30-minute average data, to reduce the high autocorrelation in the data and limit the impact of the high-frequency large variability of turbulent quantities."

- Apart from tilt correction, are data rotated into the mean wind?

  As described at the DOI of the data (included in the data availability section), data have been rotated into a geographic coordinate system. We have now also included this specification in the manuscript.

- Given the forested nature of Perdigão, has the displacement height been taken into account? Is it assured that the measurements are above the canopy layer and roughness sublayer or are the authors testing the parametrization irrespective of the PBL layer that is probed?

  To include the effect of canopy in the machine learning models, we have now added a vegetation-related feature as input to the ML algorithms:

  - the mean vegetation height $\overline{h_{\text{veg}}}$ in the upwind 1-km radius sector centered on the measurement point. Given the forested nature of the Perdigão region, we expect canopy to have an effect in triggering turbulence, especially at lower heights. To compute this variable, we use data from a lidar survey during the season of the field campaign, at a 20 m horizontal resolution.

- What is the number of data points used when all the quality criteria are satisfied?

  We have added the following sentence in Section 2.2 "After all the quality controls have been applied, a total (from all sonic anemometers) of over 284,000 30-minute average ε data remains for the analysis."

2. **Dissipation calculation**

   Given the very large array of different sonic anemometers, can the author discuss if there were any noticeable differences in the estimated dissipation rates? Is aliasing observed at tau = 0.1s for any of the datasets especially in stable conditions? Have the authors performed a quality control of the dissipation rate based on if the slope of the structure function is really 2/3 (plus/minus some uncertainty interval)?

   We agree with the reviewer that it is important to add some quality control on the dissipation rate values used in the analysis. To this regard, we have implemented the following QC based on the propagation of errors:

To account for the uncertainty in the calculation of $\epsilon$, we apply the law of combination of errors, which tracks how random errors propagate through a series of calculations (Barlow, 1989). We apply this method to equation 2 and quantify the fractional standard deviation in the $\epsilon$ estimates (Piper, 2001; Wildmann et al., 2019) as

$$\sigma_\epsilon = \frac{3}{2}\frac{\sigma_I}{I}\epsilon \tag{3}$$

where $I$ is the sample mean of $\tau^{-2/3}D_U(\tau)$, and $\sigma_I^2$ is its sample variance. To perform our analysis only on lowly-uncertain $\epsilon$ values, we discard dissipation rates characterized by $\sigma_\epsilon > 0.05$. About 3% of the data are discarded based on this criterion.

**3. Multivariate linear regression**

- The multivariate linear regression shows the worst results of the machine learning methods used. At a first glance this comes as no surprise given that the dissipation rate is not necessarily related to other variables in a linear way, despite the fact that it is commonly accepted that dissipation and TKE are strongly coupled. The authors also mention that it is due to dissipation rate spanning multiple order of magnitude more than the TKE. However, the method might be underperforming because of a different reason. Since the response variable is the logarithm of the dissipation rate, there is no reason to expect that the predictor variables should be variables themselves rather than logarithms. For example, equation (5) shows that the parametrization of dissipation is related to TKE through:

$$\epsilon = \frac{TKE^{\frac{2}{3}}}{BL_M}$$

If we now want to see how logarithm of dissipation rate is related to TKE we see that it is related to the logarithm of TKE and not to TKE itself:

$$\log_{10}\epsilon \approx \frac{2}{3}\log_{10}TKE - \log_{10}L_M$$

Indeed, plotting $\log_{10}\epsilon$ vs TKE produces a similar shape to the one observed in Fig. 8, while $\log_{10}\epsilon$ vs $\log_{10}TKE$ are linearly related.

I expect that the multi-linear regression will produce a much more significant results with better $R^2$ and less bias if the predictor variables (TKE, u*, z/L) are switched with their logarithms. In the logarithmic representation there will also be only one TKE representation necessary. I suspect the same approach will produce an even better result for random forests.

We agree with the reviewer, and thank her for pointing this out. We have now modified the set of input features used in our study, and re-done the analysis accordingly. Section 4.4 describes in detail the new set of input features used:

**4.4 Input features for machine-learning algorithms**

Given the large variability of $\epsilon$, which can span several orders of magnitude (Bodini et al., 2019b), we apply the machine-learning algorithms to predict the *logarithm* of $\epsilon$. To select the set of input features used by the learning models, we take advantage of the main findings of the observational studies on the variability of $\epsilon$ to select as inputs both atmospheric- and terrain-related variables to capture the impact of topography on atmospheric turbulence. For each variable, we calculate and use in the machine learning algorithms 30-minute average data, to reduce the high autocorrelation in the data and limit the

impact of the high-frequency large variability of turbulent quantities. We use the following input features (calculated at the same location and height as $\epsilon$) for all of the considered learning algorithms:

- wind speed (WS), which has been shown to have a moderate correlation with $\epsilon$ (Bodini et al., 2018);

- the logarithm of TKE, which is expected to have a strong connection with $\epsilon$ according to Eq. (4), calculated as

$$\log(\text{TKE}) = \log\left[\frac{1}{2}\left(\sigma_u^2 + \sigma_v^2 + \sigma_w^2\right)\right] \tag{13}$$

where the variances of the wind components are calculated over 30-minute intervals. The choice of using the *logarithm* of TKE is justified by the fact Eq. 4 suggests this quantity is linearly related to the logarithm of $\epsilon$;

- the logarithm of friction velocity $u_*$, which is calculated as

$$u_* = (\overline{u'w'}^2 + \overline{v'w'}^2)^{1/4}. \tag{14}$$

An averaging period of 30 minutes (De Franceschi and Zardi, 2003; Babić et al., 2012) has been used to apply the Reynolds decomposition and calculate average quantities and fluctuations.

- the log-modulus transformation (John and Draper, 1980) of the ratio $\zeta = z_{\text{son}}/L$, where $z_{\text{son}}$ is the height above the ground of each sonic anemometer, and $L$ is the 30-minute average Obukhov length:

$$\text{sign}(\zeta)\log(|\zeta|+1) \tag{15}$$

The use of $\zeta$ is justified within the context of the Monin Obukhov similarity theory (Monin and Obukhov, 1954). The use of the logarithm of $\zeta$ is consistent with the use of the logarithm of $\epsilon$ as target variable. Finally, the log-modulus transformation allows for the logarithm to be calculated on negative values of $\zeta$ and be continuous in zero.

- the standard deviation $\text{std}(z_{\text{terr}})$ of the terrain elevation in a 1-km radius sector centered on the measurement point (i.e., the location of the sonic anemometer). The angular extension of the sector is set equal to $\pm 30°$ from the recorded 30-minute average wind direction (an example is shown in Figure 7). While we acknowledge that some degree of arbitrariness lies in the choice of this variable to quantify the terrain influence, it represents a quantity that can easily be derived from numerical models, should our approach be implemented for practical applications, to capture the influence of upwind topography to trigger turbulence. To compute this variable, we use Shuttle Radar Topography Mission (SRTM) 1 Arc-Second Global data, at 30 m horizontal resolution.

- the mean vegetation height $\overline{h_{\text{veg}}}$ in the upwind 1-km radius sector centered on the measurement point. Given the forested nature of the Perdigão region, we expect canopy to have an effect in triggering turbulence, especially at lower heights. To compute this variable, we use data from a lidar survey during the season of the field campaign, at a 20 m horizontal resolution.

The distribution of the input features and of $\log(\epsilon)$ are shown in the Supplement.

The distribution of the input features in the Supplement have been modified accordingly.

- I miss the information on what variables were chosen by the multivariate model? The results are only presented for the random forest. With so many related variables the full model should be penalized.
We are not sure we exactly understand this comment. If the reviewer is asking about the input features used in the model, these are the same used for all three the models used in our analysis. We have specified this in Section 4.4: "We use the following input features for the three learning algorithms considered in our study:".
If the reviewer is instead asking about the model weights (i.e. the coefficients of the multivariate regression), these are not shown as they cannot be directly

determined from the nested cross validation approach followed in our analysis. Such an approach is aimed at getting the most accurate estimate of the generalization error of the learning algorithm, but will not provide a single estimate of the model weights, as more one "optimal" model is found for each nested run. Nevertheless, we are reporting a detailed analysis of the physical interpretation of the machine learning results in Section 5.2.

- Can the authors discuss more in depth their motivation for choosing the parameters they did maybe within the Monin-Obukhov framework or HOST framework for stable conditions?
  The description of the input variables now includes more comments in this sense.

4. **Influence of measurement height**
   Given that there are only a couple of towers that are 100 m high I am wondering about the representativity of these very high measurements as they will occupy only a very small fraction of the training. If one uses z/L then this influence will be normalized and will no longer be an issue, however, the authors use height of the sonic $z_{son}$ which is not normalized and therefore subject to representativity issues.
   We agree with the reviewer. We have now removed z and L from the set of input features, and used instead a variable derived from $\log(z/L)$:
   – the log-modulus transformation (John and Draper, 1980) of the ratio $\zeta = z_{son}/L$, where $z_{son}$ is the height above the ground of each sonic anemometer, and $L$ is the 30-minute average Obukhov length:

   $$\text{sign}(\zeta)\log(|\zeta|+1) \tag{15}$$

   The use of $\zeta$ is justified within the context of the Monin Obukhov similarity theory (Monin and Obukhov, 1954). The use of the logarithm of $\zeta$ is consistent with the use of the logarithm of $\epsilon$ as target variable. Finally, the log-modulus transformation allows for the logarithm to be calculated on negative values of $\zeta$ and be continuous in zero.

   In the same way I wonder about the results of Figure 6 in which the mean bias according to height of MYNN is shown. The results for lower heights will include a more varied set of conditions than for higher heights. I would find it justified to compare the bias for different heights only on the towers with similar heights (for example the two 100m towers).
   We have added some error quantification to Figure 6 to quantify the spread of the results shown at each height:

[Figure]

We have also performed the same analysis only using data from the three 100-m towers, and added a comment in the main paper and a figure in the Supplementary Information: "We obtain comparable results when computing the bias in the MYNN parameterization only for the sonic anemometers mounted on the three 100-m meteorological towers (Figure shown in the Supplement), thus confirming that the observed trend is not due to the larger variability of the conditions sampled by the more numerous sonics at lower heights. Therefore, our results show how the MYNN formulation fails in accurately representing atmospheric turbulence especially in the lowest part of the boundary layer."

[Figure]

Figure S1: Mean bias in the MYNN-parameterized $\log(\epsilon)$ at different heights, as calculated from the sonic anemometers on the three 100-m towers at Perdigão.

**5. Terrain influence**

The terrain influence in the paper is quantified through a standard deviation of the terrain within 1 km upstream of the measurement. I presume that this is because such a variable is readily available in numerical models, but this motivation is missing in the paper. On the other hand, from a physical point of view I am wondering how this variable can be justified. Given the variety of measurement heights in the dataset the flux footprint and therefore also the terrain that influences the measurement is going to vary substantially. It would

therefore be good to either motivate the choice in detail, or to present some footprint analysis which convinces us that the choice of 1 km is meaningful. I also miss the information on how this standard deviation is computed. What is the resolution of the digital elevation model used for this computation? And what is the reasoning behind using standard deviation as opposed to for example slope angle? Given the change of the footprint with height, wouldn't it be more appropriate to estimate the effect of terrain only for measurements with similar height?

In the description of the 'new' set of input features used (see answer to specific comment #2) we have added a comment on how the standard deviation of upwind terrain has been chosen as it can be easily computed from numerical models. We have also added details on the DEM dataset used to compute this variable in our analysis.

In Section 5.2, we state that "Though not negligible, the importance of topography and canopy might increase by considering different parameters that could better encapsulate their effect."

Finally, we have performed an additional analysis on the importance of the input features for the random forest prediction when single heights are considered:

"We have tested how the feature importance varies when considering several random forests, each trained and tested with data from all the sonic anemometers at a single height only, and did not find any significant variation of the importance of the considered variables in predicting ε (plot shown in the Supplement)."

[Figure]

**6. Separation according to stability**

Results of Fig 5 show very large difference in the success of the parametrization for stable and unstable stratification. Looking at the results I would say that there is visually almost no need for improvements on the unstable side. With this in mind, I wonder why the approach is then followed which lumps all the data together.

We have now added a more detailed analysis of the random forest results based on stability:

Given the large gap in the performance of the MYNN parameterization of $\epsilon$ between stable and unstable conditions, it is worth exploring how the machine learning algorithms perform in different stability conditions. To do so, we train and test two separate random forests: one using data observed in stable conditions, the other one for unstable cases. We find that both algorithms eliminate the bias observed in the MYNN scheme (Figure 9). The random forest for unstable conditions provides, on average, more accurate predictions (RMSE = 0.37, MAE = 0.28) compared to the algorithm used for stable cases (RMSE 0.44, MAE = 0.33), thus confirming the complexity in modeling atmospheric turbulence in quiescent conditions. However, when the error metrics are compared to those of the MYNN parameterization, the random forest for stable conditions provides the largest relative improvement, with a 50% reduction in MAE, while for unstable conditions the reduction is of 20%.

[Figure]

**Figure 9.** Density histogram showing the comparison, performed on the testing set, between observed and machine-learning-predicted $\epsilon$ from a random forest for stable conditions (left) and unstable conditions (right).

**7. Paper structure**
The paper structure could be improved if machine learning algorithms were introduced before the predictor variables that are used to feed these algorithms.

We have changed the structure of the paper following your feedback, and the machine learning algorithms are now presented before the input features.

**Minor points**

1. Ln 35: "; for example" should be ": for example"
   Changed.

2. Figure 1: It would be good to color the points according to the height of the tower their represent

Done:

[Figure]

3. Ln 69: "are recorded" should be "were recorded"
Changed.

4. Figure 2: given that this figure is only for presentation purposes I would suggest replacing a histogram for a bar plot which correctly represents the measurement heights. This could still be done in some meaningful increments but would not bundle 2m heights under 0 and would not have gaps for say 90 m height which does not exist
We have replaced the figure with the following:

[Figure]

We have also added the following table to make the information provided more detailed:

**Table 1.** Details on heights where sonic anemometers were mounted on the meteorological towers at the Perdigão field campaign.

| Nominal tower height | Sonic anemometer heights (m AGL) | Number of towers |
|---|---|---|
| 2 m | 2 | 1 |
| 10 m | 10 | 5 |
| | 2, 10 | 5 |
| 20 m | 10, 20 | 10 |
| | 2, 10, 20 | 6 |
| | 2, 4, 6, 8, 10, 12, 20 | 4 |
| 30 m | 10, 30 | 3 |
| | 2, 4, 6, 8, 10, 12, 20, 30 | 5 |
| 60 m | 10, 20, 30, 40, 60 | 5 |
| | 2, 4, 6, 8, 10, 12, 20, 30, 40, 60 | 1 |
| 100 m | 10, 20, 30, 40, 60, 80, 100 | 3 |
| | Total number of towers | 48 |
| | Total number of sonic anemometers | 184 |

5. Ln 75-77: it is not necessary to mention that it is a structure function of horizontal velocity twice in this sentence
   We have rephrased as follows:

   TKE dissipation rate from the sonic anemometers on the meteorological towers is calculated from the second-order structure function $D_U(\tau)$ of the horizontal velocity $U$ (Muñoz-Esparza et al., 2018):

   $$\epsilon = \frac{1}{U\tau}[aD_U(\tau)]^{3/2} \tag{1}$$

   where $\tau$ indicates the temporal increments over which the structure function is calculated, and $a = 0.52$ is the one-dimensional

6. Ln 79: the part of the sentence "is done using the temporal separation between" is not very clear. Do you mean that you calculate the dissipation rate for lags between 0.1 and 2s by assuming that this is the inertial subrange?
   We have rephrased the sentence as: "We calculate ε every 30 s, and then average values at a 30-minute resolution. At each calculation of ε, we fit experimental data to the Kolmogorov model (Kolmogorov, 1941; Frisch, 1995) using time lags separation between τ₁ = 0.1 s and τ₂ = 2 s, which represent a conservative choice to approximate the inertial subrange (Bodini et al., 2018).".

7. Ln 87: algorithms haven't been introduced yet
   See answer to specific comment #7.

8. Ln 102: this is not sensible heat flux but buoyancy flux, given that the authors mention no Schotanus correction. Also, is θ_V really virtual temperature or rather sonic temperature?
   We have corrected this sentence and stated we are using buoyancy flux. We have also specified that "$\vartheta_v$ is the virtual potential temperature (K, here approximated as the sonic temperature)".

9. Ln 103: Why are the authors using values of L to define stability ranges, when it is more common to define them through z/L, where neutral stratification has a clear meaning, whereas L is not as clearly specified?
   We have now classified atmospheric stability based on z/L instead of L: "For atmospheric stability, we classify unstable conditions as ζ = z/L < -0.02; and stable conditions as ζ > 0.02; nearly-neutral conditions as |ζ| ≤ 0.02."

10. Ln 129: Given the many profiles that exist in the data, I wonder why it is impossible to estimate the L$_T$ and L$_B$ scales. The TKE is not expected to vary so erratically to not be possible to estimate its vertical variability with an analytical function.

While we agree with the reviewer that some assumptions could be made to approximate the other two length scales, we think this is not strictly necessary in the context of our paper. To better explain this point, we have added the following comment: "The observed bias would be even larger if LM was calculated including all the contributions according to Eq. (5), and not Ls only as in our approximation. Therefore, while the approximation in Eq. (9) is major and could be eased by making assumptions on the vertical profile of TKE at Perdigão, it does not affect the conclusion of a high inaccuracy in the MYNN parameterization of ε."

We have also added to the Supplementary Information the analytical proof that our approximation determines an overestimation of LM.

11. Ln 140: so is TKE then calculated at 30s?

See answer to your specific comment #1.

12. Ln 163: What do you mean by "time stamps with missing data"? Do you mean that only those periods when all the instruments had all the values were used?

We have clarified as: "No data imputation was performed, and missing data were removed from the analysis."

13. Ln 165 – 166: What do you mean by hyperparameters? Are you referring to the ones defined in Table 1? This should be referenced here.

We have rephrased as "hyperparameters (model parameters whose values are set before the training phase and that control the learning process)".

Table 1 only shows the hyperparameters of the random forest, while the linear and polynomial regression only have one hyperparameter (i.e. the alpha parameter for Ridge regression). To make this clear, we have added the following sentence: "Before testing the models, however, it is important to avoid overfitting by setting the values of hyperparameters. Each learning algorithm has specific model-specific hyperparameters that need to be considered, as will be specified in the description of each algorithm."

14. Ln 194: Mention that Scikit-learn is a python library.

We have rephrased as "python's library Scikit-learn".

15. Ln 195: what are the variables chosen by the ridge regression?

See answer to specific comment #3.

16. Ln 223: I find this sentence not very clear. Values of what were sampled in the cross-validation search? And what do you mean by five sets of parameters?

We have clarified the sentence as: "Table 2 describes which hyperparameters we considered for the random forest algorithm. For each hyperparameters listed, we include the range of values that are randomly sampled in the cross-validation search to form the ten sets of hyperparameters used in the training phase."

17. Table 1: How were these values chosen?
For some hyperparameters, the choice of their values is constrained by the problem: for example, the maximum number of features has to be picked based on the number of features of the specific problem. For other parameters, the minimum value is often 1, while the maximum sampled values are chosen (after some empiric tests and/or past experience) to avoid allowing for a model that is complicated enough to overfit the problem.

18. Ln 232: How do you explain this "optimistic result" that using a reduced parametrization is actually beneficial to using the full one?
We have clarified what we mean by "optimistic result": "We note that, because the length scale approximation we made in calculating MYNN-predicted $\varepsilon$ led to a better agreement with the observed values compared to what would be obtained using the full MYNN parameterization, the RMSE and MAE for the MYNN case would in reality be higher than what we report here, and so the error reductions achieved with the machine-learning algorithms would even be greater than the numbers shown in the Table."

19. Ln 252: Is $R^2$ the adjusted one that takes into account the penalization for overfitting? Are all the variables statistically significant and at which p value?
To remove ambiguity and be consistent with the error metrics used throughout the paper, we have removed $R^2$ from the table.

20. Ln 265: Within Monin-Obukhov similarity theory L is not the relevant variable but z/L. The use of logarithm of (z/L) might improve the importance of this variable.
As already mentioned, we have now used a variable derived from log(z/L) as input feature for the machine learning algorithms.

*In this document, the reviewer's comments are in black, the authors' responses are in red.*

The authors thank the reviewer for their thoughtful and productive comments.

The manuscript "Can machine learning improve the model representation of TKE dissipation rate in the boundary layer for complex terrain?" by Bodini et al. provides an interesting look at using machine learning techniques to generate estimates of TKE dissipation rate and comparing those results to the approach used in the MYNN parameterization. This work should be of interest to the community and provides a useful road map for scientists wanting to apply a similar approach to other data sets. Overall, I think the manuscript will be acceptable for publication in Geoscientific Model Development after relatively minor revisions. The text is generally clearly written and straight forewarned to follow. I wonder, given that focus on data analysis rather than atmospheric model development, if the manuscript is a better fit for Atmosphere Chemistry and Physics or Atmospheric Measurement Techniques. I leave that, however, up to the editor.

Thank you for finding our work interesting and well-structured. Regarding the choice of the journal, we would like to emphasize that GMD has already published at least another paper (reference below) with a focus similar to ours, and therefore we think that adding another publication on the topic in the same journal would strengthen both papers. In addition, the focus of our work is on explaining weaknesses in MYNN parameterization and working towards a possible replacement, hence we think this fits into GMD's scope of "new methods for assessment of models, including work on developing new metrics for assessing model performance and novel ways of comparing model results with observational data".
*Leufen, L. H. and Schädler, G.: Calculating the turbulent fluxes in the atmospheric surface layer with neural networks, Geosci. Model Dev., 12, 2033–2047, https://doi.org/10.5194/gmd-12-2033-2019, 2019.*

**General comments**

- Machine learning techniques generally do not increase our physical understanding. The authors try to address this in Section 5.1 and 5.2 where additional analysis is provided. Section 5.2, however, is very brief and should be developed more to provide additional insight into the results.
  To give more importance to the physical interpretation of the machine learning results, we have now unified Sections 5.1 and 5.2 and used "Physical interpretation of machine learning results" as header.
  We have also added a new analysis on the performance of the random forest for different stability conditions – see answer to the next general comment.
  In addition, we have added more comments on the description of the partial dependence analysis, and added plots for all the input features used.
  Finally, we have performed an additional analysis on the importance of the input features for the random forest prediction when single heights are considered:
  "We have tested how the feature importance varies when considering several random forests, each trained and tested with data from all the sonic anemometers at a single height only, and did not find any significant variation of the importance of the considered variables in predicting $\varepsilon$ (plot shown in the Supplement)."

[Figure]

- In section 3, the authors show that the MYNN approach does a reasonable job in unstable conditions, but much worse when the boundary layer is statically stable. I was surprised that the authors didn't carry this analysis into the subsequent sections. It would seem natural to examine the model behavior with stability in Section 5.

  We have now added a more detailed analysis of the random forest results based on stability:

  Given the large gap in the performance of the MYNN parameterization of $\epsilon$ between stable and unstable conditions, it is worth exploring how the machine learning algorithms perform in different stability conditions. To do so, we train and test two separate random forests: one using data observed in stable conditions, the other one for unstable cases. We find that both algorithms eliminate the bias observed in the MYNN scheme (Figure 9). The random forest for unstable conditions provides, on average, more accurate predictions (RMSE = 0.37, MAE = 0.28) compared to the algorithm used for stable cases (RMSE 0.44, MAE = 0.33), thus confirming the complexity in modeling atmospheric turbulence in quiescent conditions. However, when the error metrics are compared to those of the MYNN parameterization, the random forest for stable conditions provides the largest relative improvement, with a 50% reduction in MAE, while for unstable conditions the reduction is of 20%.

[Figure]

**Figure 9.** Density histogram showing the comparison, performed on the testing set, between observed and machine-learning-predicted $\epsilon$ from a random forest for stable conditions (left) and unstable conditions (right).

**Specific comments**

1. Figure 1. I appreciate the histogram shown in Figure 2, but could you also differentiate the points in Figure 1 to indicate measurement heights? Maybe that doesn't work well if the measurements made at a single location are at several heights?
Yes, multiple sonics at several heights were installed on each tower. However, to give the reader a better idea of the distribution of the tower heights, we have changed the map to reflect this information:

[Figure]

2. Section 2.1: Can you say anything more about how the sonics are distributed on the towers? For example, how many were deployed on the 100 m tower?
We have added the following table to include more details on the measurement heights of the sonic anemometers:

**Table 1.** Details on heights where sonic anemometers were mounted on the meteorological towers at the Perdigão field campaign.

| Nominal tower height | Sonic anemometer heights (m AGL) | Number of towers |
|---|---|---|
| 2 m | 2 | 1 |
| 10 m | 10 | 5 |
| | 2, 10 | 5 |
| 20 m | 10, 20 | 10 |
| | 2, 10, 20 | 6 |
| | 2, 4, 6, 8, 10, 12, 20 | 4 |
| 30 m | 10, 30 | 3 |
| | 2, 4, 6, 8, 10, 12, 20, 30 | 5 |
| 60 m | 10, 20, 30, 40, 60 | 5 |
| | 2, 4, 6, 8, 10, 12, 20, 30, 40, 60 | 1 |
| 100 m | 10, 20, 30, 40, 60, 80, 100 | 3 |
| | Total number of towers | 48 |
| | Total number of sonic anemometers | 184 |

3. Lines 78-80: Double check this sentence, the wording seems odd.

We have rephrased the sentence as: "We calculate ε every 30 s, and then average values at a 30-minute resolution.. At each calculation of ε, we fit experimental data to the Kolmogorov model (Kolmogorov, 1941; Frisch, 1995) using time lags separation between $\tau_1 = 0.1$ s and $\tau_2 = 2$ s, which represent a conservative choice to approximate the inertial subrange (Bodini et al., 2018).".

4.  Line 101: Is the mean potential temperature computed from the sonic data or does it come from a different source?
    Yes, and we have now specified it: "$\vartheta_v$ is the virtual potential temperature (K, here approximated as the sonic temperature)".

5.  Lines 104-109: Can you point the reader to the terrain data set that was used? What was the resolution of that data set? Does that have any impact on the results?
    We have added additional details on this:

    – the standard deviation $\text{std}(z_{\text{terr}})$ of the terrain elevation in a 1-km radius sector centered on the measurement point (i.e., the location of the sonic anemometer). The angular extension of the sector is set equal to $\pm 30°$ from the recorded 30-minute average wind direction (an example is shown in Figure 7). While we acknowledge that some degree of arbitrariness lies in the choice of this variable to quantify the terrain influence, it represents a quantity that can easily be derived from numerical models, should our approach be implemented for practical applications, to capture the influence of upwind topography to trigger turbulence. To compute this variable, we use Shuttle Radar Topography Mission (SRTM) 1 Arc-Second Global data, at 30 m horizontal resolution.

    We have also included the relevant information in the Data availability section.

6.  Line 138: I agree that the length scale assumption is the best you can do given the data set that you have, but I think some additional discussion is warranted to help defend that selection. Can you argue that Ls is likely dominate near the surface?
    To better explain our approximation, and why we don't think that additional assumptions are strictly needed for our analysis, we have added the following comment: "The observed bias would be even larger if LM was calculated including all the contributions according to Eq. (5), and not Ls only as in our approximation. Therefore, while the approximation in Eq. (9) is major and could be eased by making assumptions on the vertical profile of TKE at Perdigão, it does not affect the conclusion of a high inaccuracy in the MYNN parameterization of ε."
    We have also added to the Supplementary Information the analytical proof that our approximation determines an overestimation of LM.

7.  Figure 6: You show the mean bias in Figure 6, could bars be added to indicate the standard deviation of the bias? This would help show how significant the biases are. In addition, the figure shows a decrease with height. Is this significant, or could it (at least partially) be related to the horizontal distribution of the measurements taken at different heights?
    We have added some error quantification to Figure 6 to quantify the spread of the results shown at each height:

[Figure]

We have also performed the same analysis only using data from the three 100-m towers, and added a comment in the main paper and a figure in the Supplementary Information: "We obtain comparable results when computing the bias in the MYNN parameterization only for the sonic anemometers mounted on the three 100-m meteorological towers (Figure shown in the Supplement), thus confirming that the observed trend is not due to the larger variability of the conditions sampled by the more numerous sonics at lower heights. Therefore, our results show how the MYNN formulation fails in accurately representing atmospheric turbulence especially in the lowest part of the boundary layer."

[Figure]

Figure S1: Mean bias in the MYNN-parameterized $\log(\epsilon)$ at different heights, as calculated from the sonic anemometers on the three 100-m towers at Perdigão.

8. Section 4: It would be helpful if you could include a brief discussion of why you selected these particular algorithms for this application.
We have added the following comment: "Given the proof-of-concept nature of this analysis in proving the capabilities of machine learning to improve numerical model parameterizations, we defer an exhaustive comparison of different machine-learning models to a future study, and only consider relatively simple algorithms in the present work."

9. Section 5.2: Is there a better header for this section to help the reader understand the importance of the analysis that is presented?
We have unified Sections 5.1 and 5.2 and used "Physical interpretation of machine learning results" as header.

10. Section 5.2: This section seems to end abruptly. Can you guide the reader to anything important? What additional insight is gained from the analysis? What does it tell us about what is controlling the dissipation rate at large values of wind speed and/or TKE?
See answer to general comments #1 and #2.

[revised manuscript text omitted]

---

## Referee Report (RR1)

**Review of "Can machine learning improve the model representation of TKE dissipation rate in the boundary layer for complex terrain?" by Bodini et al., for Geophysical Model Development.**

Spuriously represented TKE dissipation rates in numerical weather prediction models are known to affect simulation results, especially for complex terrain. In the presented manuscript, this problem is addressed by investigating if machine learning techniques can help to improve the representation of the TKE dissipation rate in comparison to established parameterization schemes. For this purpose, the authors first demonstrate the deficiencies of the commonly used Mellor, Yamada, Nakanishi, and Niino (MYNN) parameterization for turbulence measurements, collected at 184 sonic anemometers during a 6-week field campaign in Perdigão, Portugal. Afterwards, three different machine learning methods are trained on the same dataset and the results are compared to MYNN. The study shows that the systematic bias of MYNN under stable conditions is significantly reduces with machine learning techniques.

The study is within the scope of GMD and addresses a relevant and interesting topic for the modelling community. The manuscript is well structured and comprehensibly written. Therefore, the paper merits publication after a few corrections.

**General Comments**
- it is surprising that land use and topography have almost no impact on the random forest algorithm. This feature of the machine learning algorithm is in contradiction to the actual importance of land use and topography on turbulence in nature, as already stated in the introduction. The authors should discuss in more detail this low sensitivity and give possible reasons. For instance, by looking at Figure 7 it can be seen that all measurement sites are located within or at the borders of a valley. Does this lead to a channeling of the wind field and consequently only to two occurring wind directions (more or less) in the dataset. This would result in a low upstream variability of $h_{veg}$ and $std(z_{terr})$, possibly explaining their little impact on the random forest algorithm.

Is the impact of land use and topography also small for the other machine learning algorithms? A simple way to assess the sensitivity w.r.t. $h_{veg}$ and $std(z_{terr})$ would be to just omit them as input features and look at the effect on RMSE and MAE. Did the authors do that and if yes, what was the outcome?

- If the low impact of land use and topography on turbulence in this study is caused by a channeling effect of the wind field, the question arises how representative the results really are. Against the background of an intended implementation of machine learning techniques into numerical weather prediction models (as stated by the authors in the conclusions), it is necessary that the method can be applied on a variety of different land use and topography conditions. The authors should therefore discuss in a bit more detail than they currently do in the conclusions how to achieve this. What are e.g. the data requirements that need to be fulfilled by other measurement datasets to account for the impact of different land use and topography conditions?

Furthermore, how would one incorporate the results of the machine learning algorithms in a numerical weather prediction model? In their reply to reviewer #1 the authors say that the model weights cannot be directly determined – but isn' t that just what one would need?

**Specific Comments:**
Lines 48 and 424: cite the accepted paper (Leufen & Schädler, 2019)

Lines 58, 105 and 435: change 'Nakanish' to 'Nakanishi'

Line 200 (Eq. 12): I guess there should be an n as upper limit in the summation over k.

Line 319: change 'seems' to 'seem'.

Line 330: omit 'ultrasimple'

Figures 3, 4, 8 and 9: I don't think 'density histogram' is the appropriate name for this kind of scatter plot.

---

## Author Response (AR2)

*In this document, the reviewer's comments are in black, the authors' responses are in red.*

The authors thank the reviewer for their second round of comments.

General comments
The authors have addressed all of my comments and have significantly improved the paper. I suggest paper to be accepted for publication after correcting the following minor technical issues.

Minor issues:
Ln 93: "lowly-uncertain" would sound better as " with lower uncertainty" . Changed.
Ln 153: "scaled and normalized" – I missed this in the first review would maybe you can add a sentence on how this was actually done? We have added "the data were scaled and normalized by removing the mean and scaling to unit variance".
Ln 204: please add that this is a python library. Added.
Ln 298 – 301: In the revised manuscript Obukhov length L is no longer a hyperparameter. I assume that the authors meant z/L, so this should be corrected. Corrected.

*In this document, the reviewer's comments are in black, the authors' responses are in red.*

The authors thank the reviewer for their thoughtful and productive comments.

Spuriously represented TKE dissipation rates in numerical weather prediction models are known to affect simulation results, especially for complex terrain. In the presented manuscript, this problem is addressed by investigating if machine learning techniques can help to improve the representation of the TKE dissipation rate in comparison to established parameterization schemes. For this purpose, the authors first demonstrate the deficiencies of the commonly used Mellor, Yamada, Nakanishi, and Niino (MYNN) parameterization for turbulence measurements, collected at 184 sonic anemometers during a 6-week field campaign in Perdigão, Portugal. Afterwards, three different machine learning methods are trained on the same dataset and the results are compared to MYNN. The study shows that the systematic bias of MYNN under stable conditions is significantly reduces with machine learning techniques.

The study is within the scope of GMD and addresses a relevant and interesting topic for the modelling community. The manuscript is well structured and comprehensibly written. Therefore, the paper merits publication after a few corrections.

**General Comments**

- it is surprising that land use and topography have almost no impact on the random forest algorithm. This feature of the machine learning algorithm is in contradiction to the actual importance of land use and topography on turbulence in nature, as already stated in the introduction. The authors should discuss in more detail this low sensitivity and give possible reasons. For instance, by looking at Figure 7 it can be seen that all measurement sites are located within or at the borders of a valley. Does this lead to a channeling of the wind field and consequently only to two occurring wind directions (more or less) in the dataset. This would result in a low upstream variability of hveg and std(zterr), possibly explaining their little impact on the random forest algorithm.

    Is the impact of land use and topography also small for the other machine learning algorithms? A simple way to assess the sensitivity w.r.t. hveg and std(zterr) would be to just omit them as input features and look at the effect on RMSE and MAE. Did the authors do that and if yes, what was the outcome?

    We agree with the reviewer that topography has an important impact on atmospheric turbulence, as we have mentioned in the introduction of our manuscript. On the other hand, capturing this impact with a single parameter can be a challenging task. The fact the standard deviation of upstream terrain elevation does not have a large direct importance in the random forest analysis could be due to the definition of the chosen parameter itself (as mentioned in the manuscript), or due to more complex inter-connections between different variables considered in the analysis. For example, we can imagine topography to have an impact on TKE itself, which is in fact the feature with largest importance in the random forest analysis. Therefore, the importance of topography might be hidden and incorporated in the one of TKE. We have added a comment on this in the manuscript: "Also, the impact of topography and canopy might be hidden as it could be already incorporated in the variability of parameters with larger relative importance, such as TKE."

The wind roses in Fernando et al. 2019 BAMS show how the wind direction regimes are quite variable across the considered domain, with potential channeling in the valley, while the towers on the ridges show a more varied set of wind directions.

Finally, we have also tested the importance of these features in the other considered learning algorithms as proposed by the reviewer, and found that they do not determine a large reduction in MAE or RMSE, which is consistent with the low importance found for the random forest.

- If the low impact of land use and topography on turbulence in this study is caused by a channeling effect of the wind field, the question arises how representative the results really are. Against the background of an intended implementation of machine learning techniques into numerical weather prediction models (as stated by the authors in the conclusions), it is necessary that the method can be applied on a variety of different land use and topography conditions. The authors should therefore discuss in a bit more detail than they currently do in the conclusions how to achieve this. What are e.g. the data requirements that need to be fulfilled by other measurement datasets to account for the impact of different land use and topography conditions?

  Furthermore, how would one incorporate the results of the machine learning algorithms in a numerical weather prediction model? In their reply to reviewer #1 the authors say that the model weights cannot be directly determined – but isn't that just what one would need?

  We have added additional considerations to the final paragraph of the Conclusions: "Finally, the learning algorithms developed here would need to be tested using data from different field experiments, to understand whether the results obtained in this study can be generalized everywhere. Data collected in flat terrain and offshore would likely need to be considered to create a more universal model to predict dissipation in various terrains. Once the performance of a machine-learning representation of $\epsilon$ has been accurately tested, its implementation in numerical weather prediction models, such as the Weather Research and Forecasting model, should be achieved."

  The nested cross validation adopted in our analysis does not provide model weights, as the nested setup leads to multiple algorithms being chosen in the cross validation. As mentioned in the manuscript, this approach was followed as the main goal of the paper is to provide the best estimate of the ML generalization error, rather than to suggest a model for direct implementation in numerical models. When a more universal ML model is going to be implemented in numerical models, a different cross validation approach will be used, so that model weights will be easily determinable.

**Specific Comments:**

- Lines 48 and 424: cite the accepted paper (Leufen & Schädler, 2019) Changed.
- Lines 58, 105 and 435: change 'Nakanish' to 'Nakanishi' Corrected.
- Line 200 (Eq. 12): I guess there should be an n as upper limit in the summation over k. Thanks for catching this – changed.
- Line 319: change 'seems' to 'seem'. Corrected.
- Line 330: omit 'ultrasimple' Changed.
- Figures 3, 4, 8 and 9: I don't think 'density histogram' is the appropriate name for this kind of scatter plot. We now refer to these plots simply as scatter plots.

[revised manuscript text omitted]
. Also, the impact of topography and canopy might be hidden as it could be already incorporated in the variability of parameters with larger relative importance, such as TKE. We have tested how the feature importance varies when considering several random forests, each trained and tested with data from all the sonic anemometers at a single height only, and did not find any significant variation of the importance of the considered variables in predicting $\epsilon$ (plot shown in the Supplement).

Finally, to assess the dependence of TKE dissipation rate on the individual features considered in this study, Figure 10 shows partial dependence plots for the input features considered in the analysis. These are obtained, for each input feature, by applying the machine-learning algorithm (here, random forests) multiple times with the other feature variables constant (at their means) while varying the target input feature and measuring the effect on the response variable (here, $\log(\epsilon)$). In each plot, the values on the y-axes have not been normalized, so that large ranges show a strong dependence of $\log(\epsilon)$ on the feature, whereas small ranges indicate weaker dependence. The strong relationship between $\epsilon$ and TKE is confirmed, as the range shown on its y-axis is the largest among all features. As TKE increases, so does $\epsilon$. A similar trend, tough with a weaker influence, emerges when considering the dependence of $\epsilon$ on friction velocity. The relationship between $\epsilon$ and wind speed shows a less clear trend, and with a weaker dependence: $\epsilon$ increases for 30-minute average wind speeds up to $\sim$2 m s$^{-1}$, and then decreases for stronger wind speed values. A more distinct trend could emerge when considering data averaged at shorter time periods. The dependence between TKE dissipation and atmospheric stability shows a moderate impact, with stable conditions (positive values of the considered metric) showing smaller $\epsilon$ values compared to unstable cases (negative values of the considered metric). Interestingly, the largest $\epsilon$ values  seem to be connected to neutral cases. Finally, both terrain elevation and vegetation height show weak impact on determining the values of $\epsilon$, as quantified by the narrow range of values sampled on the y-axis for these two variables.

**6 Conclusions**

Despite turbulence being a fundamental quantity for the development of multiple phenomena in the atmospheric boundary layer, the current representations of TKE dissipation rate ($\epsilon$) in numerical weather prediction models suffer from large inaccuracies. In this study, we quantified the error introduced in the MYNN parameterization of $\epsilon$ by comparing predicted and observed values of $\epsilon$ from 184 sonic anemometers from 6 weeks of observations at the Perdigão field campaign. A large positive bias (average +12% in logarithmic space, +47% in natural space) emerges, with larger errors found in atmospheric stable conditions. The need for a more accurate representation of $\epsilon$ is therefore clearly demonstrated.

The results of this study show how machine learning can provide new ways to successfully represent TKE dissipation rate from a set of atmospheric and topographic parameters. Even  simple models such as a multivariate linear regression can provide an improved representation of $\epsilon$ compared to the current MYNN parameterization. More sophisticated algorithms, such as a random forest approach, lead to the largest benefits, with over a 35% reduction in the average error introduced in the parameterization of $\epsilon$, and eliminate the large bias found in it, for the Perdigão field campaign. When considering stable

[Figure]

**Figure 10.** Partial dependence plots for the input features used in the analysis. Distributions of the considered features are shown in the background.

conditions only, the reduction in average error reaches 50%. Although the generalization gap between the universal nature of the MYNN parameterization of $\epsilon$ and the campaign-specific training and testing of the machine-learning models needs to be acknowledged, the results of this study can be considered as a proof of concept of the potentialities of machine-learning-based representations of complex atmospheric processes.

340     Multiple opportunities exist to extend the work presented here. In the future, additional learning algorithms, such as support vector machines and extremely randomized trees, should be considered. Deep learning methods, such as recurrent neural networks, and specifically long-short term memory, which are well-suited for time-series-based problems, could also be considered to obtain a more complete overview of the capabilities of machine-learning techniques for improving numerical representations of $\epsilon$. Moreover, additional input features could be added to the learning algorithms to possibly identify additional variables with

345 a large impact on atmospheric turbulence. Finally, the learning algorithms developed here would need to be tested using data from different field experiments, to understand whether the results obtained in this study can be generalized everywhere. Data collected in flat terrain and offshore would likely need to be considered to create a more universal model to predict dissipation in various terrains. 
[revised manuscript text omitted]